# Innate scavenger receptor-A regulates adaptive T helper cell responses to pathogen infection

Zhipeng Xu[1], Lei Xu[1], Wei Li[1], Xin Jin[1], Xian Song[1], Xiaojun Chen[1], Jifeng Zhu[1], Sha Zhou[1], Yong Li[1], Weiwei Zhang[1], Xiaoxiao Dong[1], Xiaowei Yang[1], Feng Liu[1], Hui Bai[2], Qi Chen[2] & Chuan Su[1]

The pattern recognition receptor (PRR) scavenger receptor class A (SR-A) has an important function in the pathogenesis of non-infectious diseases and in innate immune responses to pathogen infections. However, little is known about the role of SR-A in the host adaptive immune responses to pathogen infection. Here we show with mouse models of helminth *Schistosoma japonicum* infection and heat-inactivated *Mycobacterium tuberculosis* stimulation that SR-A is regulated by pathogens and suppresses IRF5 nuclear translocation by direct interaction. Reduced abundance of nuclear IRF5 shifts macrophage polarization from M1 towards M2, which subsequently switches T-helper responses from type 1 to type 2. Our study identifies a role for SR-A as an innate PRR in regulating adaptive immune responses.

[1] Jiangsu Province Key Laboratory of Modern Pathogen Biology, Department of Pathogen Biology and Immunology, Nanjing Medical University, Nanjing, Jiangsu 210029, China. [2] Atherosclerosis Research Center, Key Laboratory of Cardiovascular Disease and Molecular Intervention, Nanjing Medical University, Nanjing, Jiangsu 210029, China. Correspondence and requests for materials should be addressed to X.C. (email: chenxiaojun0201@126.com) or to C.S. (email: chuansu@njmu.edu.cn).

Class A scavenger receptor (SR-A), also called macrophage scavenger receptor (MSR), is expressed primarily on the plasma membrane or on the Golgi apparatus of macrophages[1,2]. SR-A has an important function in many macrophage-associated biological processes (for example, adhesion and phagocytosis) and pathological conditions resulting from non-infectious diseases (for example, atherosclerosis and Alzheimer's disease)[3,4]. In addition, on pathogen infection, SR-A initiates innate immune responses in the host not only by recognizing diverse pathogen-associated molecular patterns, including lipopolysaccharide (LPS), lipoteichoic acid, bacterial cytosine guanosine dinucleotide DNA and double-stranded RNA[5], but also acting as a phagocytic receptor that mediates direct phagocytosis of various pathogenic bacteria, such as *Listeria monocytogenes*, *Staphylococcus aureus*, *Streptococcus pyogenes* and *Neisseria meningitides*[6–9], or antigens such as tumour heat shock proteins[10].

The mammalian immune system fights infection through the cooperative actions of innate and adaptive immunity. The magnitude and quality of the adaptive immune response, which provides the host with long-term protection against pathogens, are controlled by the innate immune recognition to infection[11]. Innate immune cells sense the presence of pathogen infection during the early phase of the immune response through innate pattern recognition receptors, such as Toll-like receptors, scavenger receptors and NOD-like receptors[5,12]. SR-A-deficient mice develop more robust CD4+ T-cell responses after ovalbumin immunization[13], which suggests a possible role of SR-A in macrophages to limit the adaptive immune activation; however, how innate SR-A controls adaptive immunity during pathogen infections is unclear.

SR-A is constitutively expressed by most macrophages, cells that have a major function as sentinels of infection or as mediators that shape the adaptive immune response[14–16]. Depending on the microenvironment, macrophages can be polarized into classically activated macrophages (M1 macrophages) and alternatively activated macrophages (M2 macrophages)[17]. M1 macrophages highly express major histocompatibility complex II (MHC class II), CD80, CD86 and CD16/32 and are able to secrete proinflammatory cytokines. Tumour necrosis factor (TNF), IL-6, IL-12 and the chemokines CXCL9, CXCL10 and CXCL11 secreted by M1 macrophages are essential for clearing bacterial, viral or fungal infections and can cause tissue damage[18]. By contrast, M2 macrophages highly express arginase-1(Arg-1), mannose receptor (CD206), anti-inflammatory factors (IL-10) and chemokines CCL17 and CCL22, which are involved in anti-parasitic functions, tissue repair and remodelling, angiogenesis and metabolic responses[18–20]. Previous study has shown that the production of cytokines, including IL-6 and TNF, by macrophages from SR-A-deficient mice is increased by myocardial infarction[21], suggesting that SR-A may inhibit M1 macrophage polarization. However, whether and how SR-A regulates macrophage polarization and adaptive immune responses are unknown.

In this study, we demonstrate that pathogens regulate innate expression of SR-A in macrophages, thus regulating macrophage polarization and altering adaptive Th cell responses by interacting with cytoplasmic interferon-regulatory factor 5 (IRF5) and inhibiting IRF5 nuclear translocation. Our data suggest an important SR-A-mediated macrophage/Th cell axis in host immunity against pathogen infections.

## Results

**Attenuated schistosomiasis japonica in SR-A-deficient mice.** Although similar infectious burdens revealed by adult worm pairs and tissue egg counts in the liver were observed in wild-type (WT) and SR-A-deficient mice (Supplementary Fig. 1a,b), the SR-A-deficient mice displayed an accelerated time to death after *Schistosoma japonicum* infection (Fig. 1a). Results showed that the granuloma size and fibrosis in the liver and intestine (Fig. 1b–f and Supplementary Fig. 1c) and serum aspartate aminotransferase (AST) and alanine transaminase (ALT) (Fig. 1g) were lower in SR-A-deficient mice than those in WT mice on *S. japonicum* infection, while the integrity of the gut epithelium was more seriously damaged (Fig. 1f), and accompanied by the development of lethal endotoxemia (Supplementary Fig. 1d) in *S. japonicun*-infected SR-A-deficient mice. These results suggest that SR-A may participate in immunopathology in *S. japonicum*-infected mice.

**SR-A deficiency results in increased Th1 responses.** Th1 responses, which may contribute to suppress immunopathology mainly by production of IFN-γ (ref. 22), were significantly enhanced in the liver, lymph node and spleen of SR-A-deficient mice 8 weeks post *S. japonicum* infection (Fig. 2a and Supplementary Fig. 2a). However, Th2 responses, with their complex and often counteracting factors, resulting in a net upregulation of liver pathology[23,24], were significantly reduced in the liver of *S. japonicum*-infected SR-A-deficient mice (Fig. 2b and Supplementary Fig. 2b). Consistent with the *in vivo* data, *in vitro* treatment of splenocytes from SR-A-deficient mice with schistosome egg antigen (SEA), a well-known stimulus of Th2 responses, which contains complex components[25], resulted in more marked increase of Th1 cells (Fig. 2c and Supplementary Fig. 2c) but not of Th2 cells (Fig. 2d and Supplementary Fig. 2d). However, there was no significant difference in Th17, Tregs and Tfh cells, which are involved in liver pathology[26–28], between infected WT and SR-A-deficient mice (Supplementary Fig. 2e–g). In addition, immunization with bacterial product heat-inactivated *Mycobacterium tuberculosis* resulted in much stronger Th1 responses in SR-A-deficient mice (Supplementary Fig. 2h). Together, these results indicate that SR-A deficiency leads to a shift towards Th1 response during infection and inflammation.

**SR-A deficiency preferentially results in M1 polarization.** Given that macrophages are the main cellular constituents of granulomas[29] and important regulators in schistosomiasis[30,31], we investigated whether SR-A regulates the macrophage polarization in *S. japonicum*-infected mouse model. CD16/32 expression was increased but CD206 (Fig. 3a,b) expression were decreased in hepatic macrophages of SR-A-deficient mice compared to those of WT mice after *S. japonicum* infection. In addition, hepatic macrophages of SR-A-deficient mice displayed enhanced expression of M1-associated genes (*iNOS*, *TNF*, *IL-6*, *CXCL9*, *CXCL10* and *CXCL11*) and deceased expression of M2-associated genes (*Arg-1*, *IL-10*, *CCL17*, *CCL22* and *Ym-1*) compared to macrophages of WT mice post *S. japonicum* infection (Fig. 3c). Consistently, *in vitro* SEA-stimulated macrophages from SR-A-deficient mice produced significantly more M1-related but less M2-related genes (Fig. 3d). Interestingly, more CXCL10 was expressed by SR-A-deficient macrophages *in vivo* than *in vitro*, probably due to the lack of TNF, IFN-γ and endotoxin *in vitro* (Supplementary Fig. 1e)[32,33].

Similarly, compared to WT macrophages, the SR-A-deficient macrophages expressed more CD16/32 but less CD206 when mice were immunized by heat-inactivated *M. tuberculosis* (Supplementary Fig. 3a).

M1 macrophage polarization has been reported to be associated with high levels of antigen-presenting capacity, due to the presence of MHC II and costimulatory molecules[17]. Intriguingly, our results showed that peritoneal macrophages from SR-A-deficient mice had a more significant upregulation of

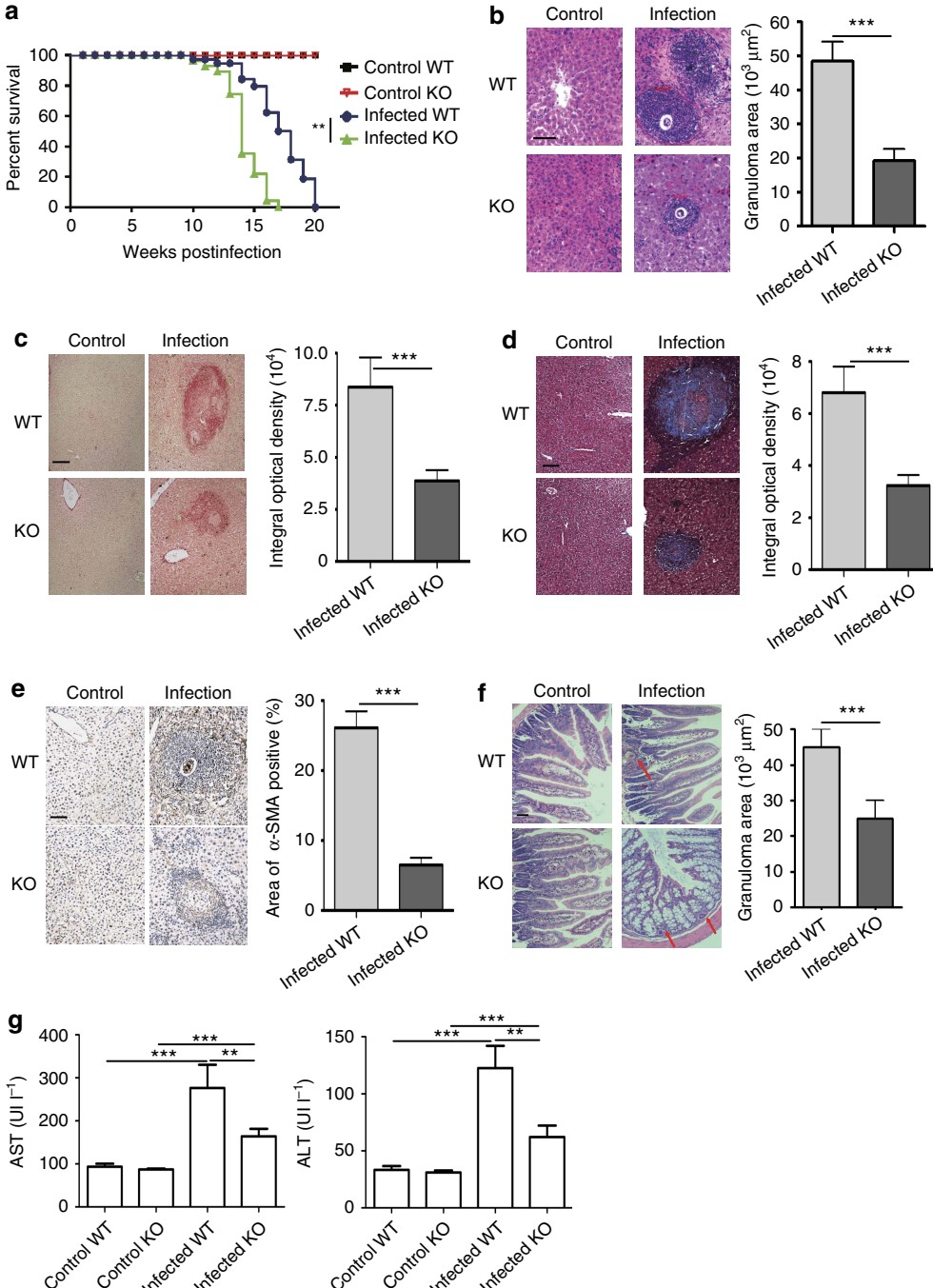

**Figure 1 | SR-A deficiency alleviates survival and pathology after *S. japonicum* infection.** (**a**) Survivals within 20 weeks post infection were calculated using Kaplan–Meier survival curves with log-rank test and $\chi^2$ statistical analysis, data are expressed as the mean ± s.d. of 24 mice from two independent experiments. (**b**,**f**) Paraffin-embedded liver sections stained with H&E. Scale bars, 100 μm. For each mouse, the sizes of 30 liver granulomas (**b**) or 10 intestine granulomas (**f**) around single eggs were quantified with AxioVision Rel 4.7. Paraffin-embedded sections were stained with sirius red (**c**), Masson (**d**) or α-SMA (**e**). Scale bars, 100 μm; The mean optical density of collagen fibres by sirius red, Masson or α-SMA staining was digitized and analysed by using Image-Pro Plus software. (**g**) Levels of serum ALT/AST were determined. Data are expressed as the mean ± s.d. of 12 mice for each group in one representative experiment. All experiments were repeated twice (**f**) or three times (**b**–**e**,**g**) with similar results, ***$P < 0.001$, **$P < 0.01$, *$P < 0.05$ (Student's *t*-test (**b**–**f**) or ANOVA/LSD (**g**).

MHC II and costimulatory molecules, including CD80 and CD86 after SEA stimulation (Supplementary Fig. 3b).

Taken together, these results suggest that SR-A deficiency leads to a skewed M1-polarized phenotype of macrophages.

**SR-A drives M2 and favours Th2 polarization.** To investigate the contribution of SR-A and macrophage polarization to

CD4+ T-cell differentiation, peritoneal macrophages were purified from *S. japonicum*-infected WT or SR-A-deficient mice (M2 or M1 dominate, respectively, Supplementary Fig. 4a) and adoptively transferred into normal WT mice. Results showed that the adoptive transfer of macrophages from infected SR-A-deficient mice with an M1-polarized phenotype (Supplementary Fig. 4a) resulted in significantly higher Th1

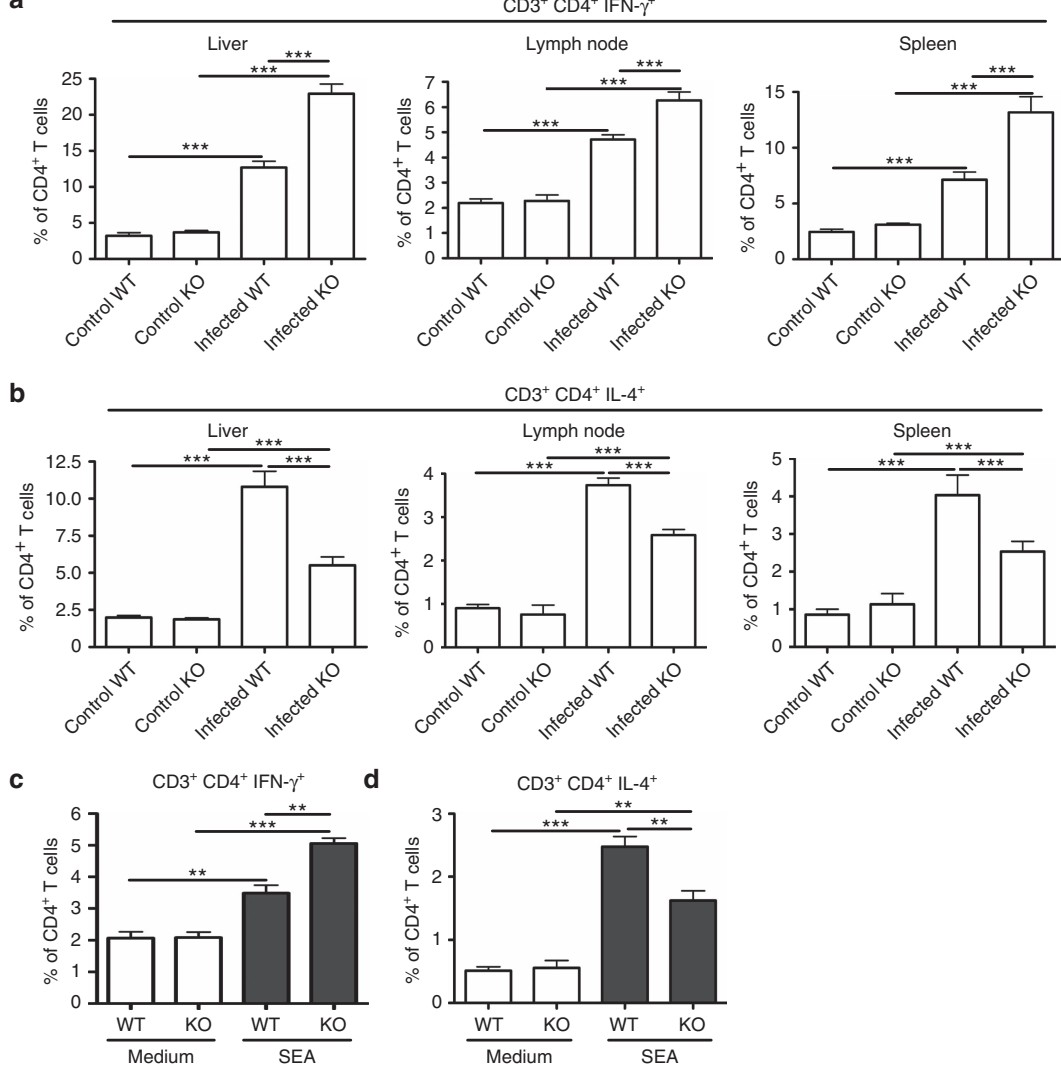

**Figure 2 | SR-A deficiency promotes Th1 response after *S. japonicum* infection.** Single-cell suspensions of mouse livers, mesenteric LN and spleens from WT and SR-A-deficient mice infected with or without *S. japonicum* were prepared. Cells were stained with CD3-APC, CD4-FITC and then intracellularly stained with PE-conjugated antibodies against IFN-γ or IL-4 for FACS analysis of CD3$^+$CD4$^+$IFN-γ$^+$ (Th1) (**a**) or CD3$^+$CD4$^+$IL-4$^+$ (Th2) (**b**) cells, respectively. Data shown are gated on CD3$^+$CD4$^+$ cells. Data are expressed as the mean ± s.d. of 12 mice for each group from one experiment representative of three independent experiments. Splenocytes from WT or SR-A-deficient mice stimulated with or without SEA, and the proportion of Th1 (**c**) or Th2 (**d**) cells in CD4$^+$ T cells was analysed by FACS. Data are expressed as the mean ± s.d. from three independent experiments, ***$P < 0.001$, **$P < 0.01$, *$P < 0.05$ (ANOVA/LSD).

responses in recipient mice (Fig. 4a and Supplementary Fig. 4b).

Next, macrophages purified from *S. japonicum*-infected WT or SR-A-deficient mice were *in vitro* co-cultured with WT CD4$^+$ T cells. Results showed that CD4$^+$ T cells were preferentially differentiated towards Th1 cells when exposed to SR-A-deficient macrophages but not WT macrophages (Fig. 4b and Supplementary Fig. 4c). In addition, *in vitro* treatment of SR-A-deficient macrophages derived from uninfected mice with SEA induced Th1 cell differentiation. In contrast, WT macrophages favoured Th2 responses in the SEA-stimulated co-culture system (Fig. 4c and Supplementary Fig. 4d). Furthermore, our data showed that the deficiency of SR-A in DC did not result in significant changes in both DC responses and Th1/Th2 induction (Supplementary Fig. 4e–g).

Next, we purified macrophages from normal WT and SR-A-deficient mice, and adoptive transfer to infected WT or SR-A-deficient mice, respectively. Results showed that transferring

SR-A-deficient macrophages to infected WT mice can significantly decrease the granuloma size, accompanied by increased Th1 response and decreased Th2 response. However, transferring of WT macrophages to *S. japonicum*-infected SR-A-deficient mice had no effect on liver egg counts but increased the granuloma size (Fig. 4d,e and Supplementary Fig. 4h), accompanied by increased Th2 response and decreased Th1 response (Fig. 4f,g and Supplementary Fig. 4i). These data suggested that the role of macrophage in the regulation of CD4$^+$ T-cell differentiation and granuloma formation during *S. japonicum* infection is SR-A intrinsic.

To further determine the contribution of SR-A to the polarization of macrophages and differentiation of CD4$^+$ T cells, siRNA was used to inhibit SR-A expression in bone marrow-derived macrophages (BMDMs). Result showed that the level of SR-A protein was significantly decreased in SR-A–siRNA-treated BMDMs (Supplementary Fig. 5a). As a result, SR-A-silenced BMDMs showed a significant increased M1 phenotype

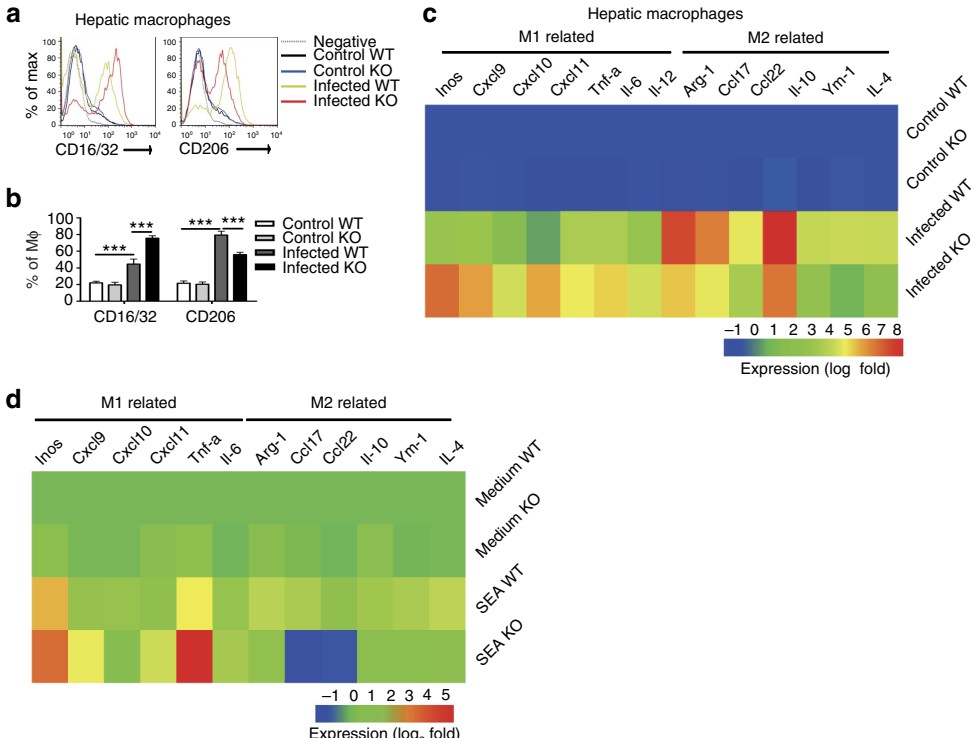

**Figure 3 | SR-A deficiency promotes M1 polarization after *S. japonicum* infection.** FACS analysis of proportion of CD16/32$^+$ or CD206$^+$ on hepatic macrophages (**a**,**b**) from WT or SR-A-deficient mice infected with or without *S. japonicum*. Data are expressed as the mean ± s.d. of 12 mice per group, and are representative of one typical experiment out of three, ***$P<0.001$, **$P<0.01$ (ANOVA/LSD). (**c**) Hepatic macrophages were purified by Flow sorting from normal or *S. japonicum*-infected WT or SR-A-deficient mice at 8 weeks post infection or (**d**) peritoneal macrophages from WT or SR-A-deficient mice were purified and stimulation with SEA. Expression of M1- or M2-related genes was evaluated by real-time PCR. Transcript levels for each gene in macrophages are expressed as fold change over transcript levels in macrophages from control WT mice or WT macrophages treated with medium, respectively. Results are presented as higher (red) or lower (blue) expression after infection or stimulation (key (below), log$_2$ fold change). Data are from three independent experiments and expressed as the mean of each group from one representative experiment.

by upregulating CD16/32 expression after SEA stimulation (Supplementary Fig. 5b), and were more efficient in driving Th1 cell differentiation when co-cultured with CD4$^+$ T cells (Fig. 4h and Supplementary Fig. 5c). These data suggest that immune dysregulation caused by SR-A deficiency is macrophage cell intrinsic and is less likely caused by the developmental defects of SR-A complete knockout. Furthermore, overexpression of SR-A in BMDMs by viral transduction profoundly attenuated SEA-induced M1 polarization (Supplementary Fig. 5d,e), resulting in enhanced Th2 differentiation when co-cultured with CD4$^+$ T cells (Fig. 4i and Supplementary Fig. 5f).

The cytokine milieu is crucial for Th cell differentiation. IL-12 has been shown to play a key role in Th1 differentiation, while IL-4 is critical for Th2 development[34]. Consistently, the expression of IL-12 both in serum and within granulomas was increased in *S. japonicum*-infected SR-A-deficient mice, while IL-4 decreased significantly (Supplementary Fig. 5g). Compared to WT macrophages, SEA-stimulated SR-A-deficient macrophages produced significantly more IL-12 but less IL-4 (Fig. 4j and Supplementary Fig. 5h), suggesting that SR-A regulates Th1/Th2 cell differentiation by modulating cytokine expression in macrophages. To further determine the cell-intrinsic role of SR-A in the regulation of IL-12/IL-4 cytokines, siRNA was used for silencing SR-A in WT peritoneal macrophages. Results showed that the mRNA level of IL-12 was increased in SR-A-siRNA-treated WT macrophages after SEA stimulation, while IL-4 decreased significantly (Fig. 4k,l). On the contrary, overexpression of SR-A in WT peritoneal macrophages profoundly attenuated SEA-induced IL-12 level and enhanced IL-4 level (Fig. 4m,n).

Together, these data indicate that SR-A suppresses M1 but promotes M2 polarization of macrophages, and subsequently downregulates IL-12p35 but upregulates IL-4 expression to regulate Th1 or Th2 differentiation, respectively.

**SR-A promotes M2 and Th2 polarization by regulating IRF5.** Transcription factor IRF5 has an important function in the induction of pro-inflammatory cytokines that contribute to the plasticity and polarization of macrophages to an M1 pheno-type[18,35]. As shown in Fig. 5a, the level of IRF5 was significantly higher in SR-A-deficient macrophages from *S. japonicum*-infected mice or after *in vitro* stimulation macrophages by antigens (SEA) (Fig. 5b). However, the level of either total or nuclear IRF4, which has also been linked to M2 macrophage polarization during helminth infection[19], showed no significant difference between WT and SR-A-deficient macrophages (Supplementary Fig. 6a,b).

To determine the possible function of IRF5 in SR-A-regulated macrophage polarization and CD4$^+$ T-cell differentiation, macrophages from *S. japonicum*-infected WT or SR-A-deficient mice were transduced with IRF5–shRNA, and then co-cultured with CD4$^+$ T cells in the presence of SEA *in vitro*. Results showed that knockdown of IRF5 resulted in significantly reduced expression of CD16/32 (Supplementary Fig. 6c) and IL-12p35 (Fig. 5c) in SR-A-deficient macrophages. Accordingly, macrophages with IRF5 knockdown significantly reduced Th1 cell differentiation when co-cultured with WT CD4$^+$ T cells (Fig. 5e and Supplementary Fig. 6d). On the contrary, silencing of IRF5

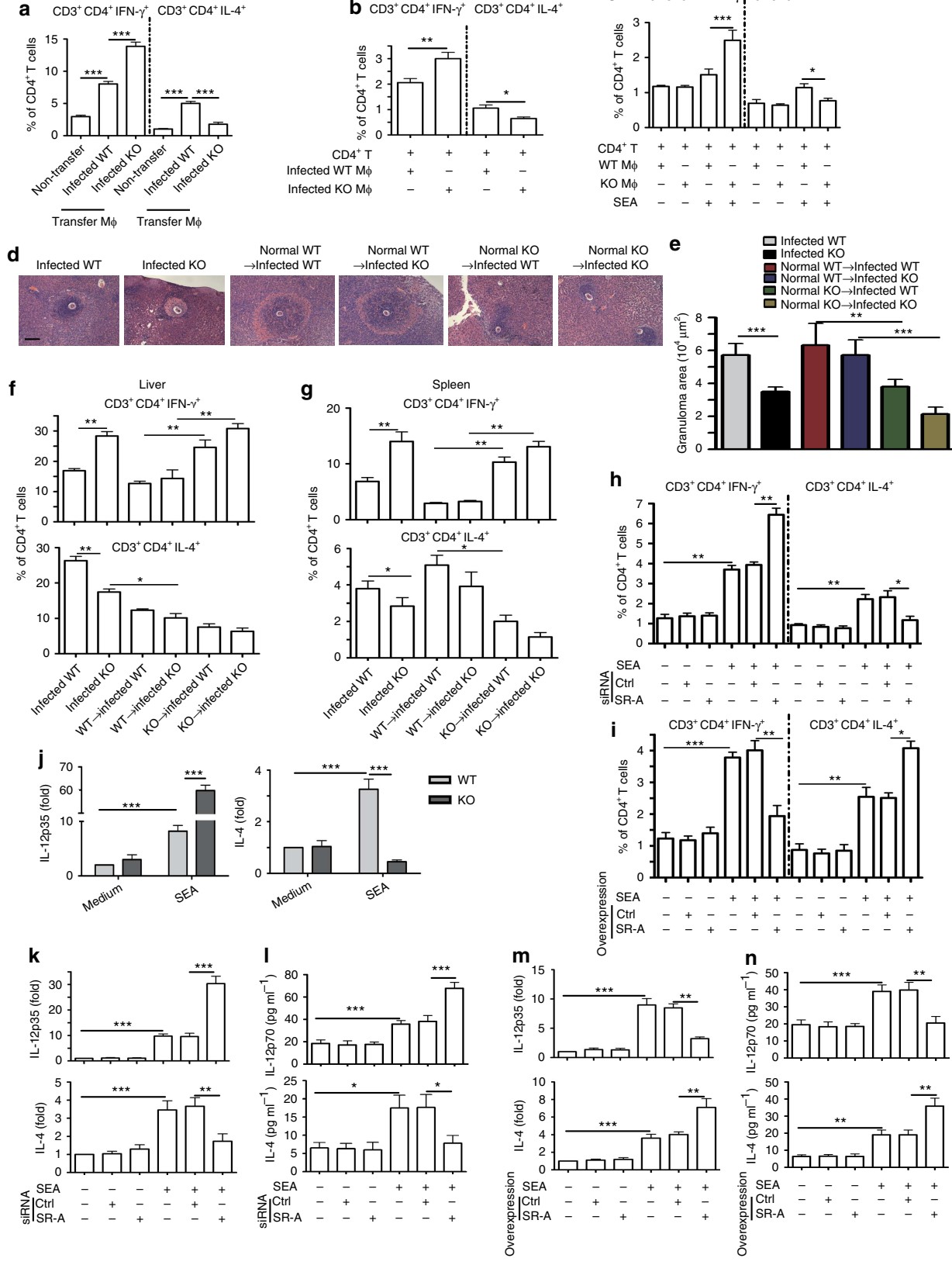

expression resulted in significantly increased CD206 (Supplementary Fig. 6c) and IL-4 (Fig. 5d) in SR-A-deficient macrophages, as well as significantly decreased Th2 cell differentiation (Fig. 5e and Supplementary Fig. 6d). Together, our data indicate that SR-A regulates the expression of IRF5, which is crucial for M1 polarization and subsequent induction of Th1 cell differentiation.

**M2 and Th2 polarization rely on SR-A and IRF5 interaction.** Considering that IRF5 is mainly expressed in monocytes, but not in T cells[36], we thus injected mice with Lentivirus-IRF5-shRNA to further determine the role of IRF5 in SR-A-regulated CD4$^+$ T-cell differentiation and liver pathology during *S. japonicum* infection (Supplementary Fig. 7a). The expression of IRF5 in macrophages of WT mice infected with Lv-IRF5-shRNA was inhibited, compared to that in PBS-treated or control shRNA-infected groups (Supplementary Fig. 7b). Knockdown of IRF5 in SR-A-deficient mice resulted in significant inhibition of M1 polarization but facilitated M2 polarization than control shRNA-infected mice (Supplementary Fig. 7c,d). Meanwhile, knockdown of IRF5 in SR-A-deficient mice also inhibited Th1 responses but promoted Th2 responses (Supplementary Fig. 7e–g). Furthermore, liver pathology (Fig. 6a,b) and liver function (AST and ALT) (Fig. 6c) in SR-A-deficient mice were restored when mice are injected Lv-IRF5-shRNA. Together, these results demonstrate that M1 polarization and Th1 response in *S. japonicum*-infected SR-A-deficient mice are mainly due to the increased expression of IRF5.

Given that IRF5 function is dependent on its nuclear translocation[37], we next assessed the effect of SR-A on the nuclear translocation of IRF5. Results showed that after infection with *S. japonicum*, there was more IRF5 in the nuclei of SR-A-deficient macrophages than WT cells (Fig. 6d,e and Supplementary Fig. 8a). Consistently, SR-A-deficient macrophages also had more IRF5 in the nuclei after stimulation with SEA *in vitro* (Supplementary Fig. 8b).

To determine whether SR-A could interact with IRF5, we first used the STRING database (http://string.embl.de/) to predict the possible interaction of SR-A and IRF5, and found that there is a possibility of two protein interactions (Supplementary Fig. 9). In addition, confocal microscopy was used to determine the co-expression of SR-A and IRF5 in RAW264.7 cell line. Result showed a co-localization of SR-A and IRF5 in the cytoplasm of RAW264.7 cells (indicated by the yellow overlap in Fig. 6f). Co-immunoprecipitation assays showed that IRF5 binding to endogenous SR-A could be detected in the cytoplasmic fractions (Fig. 6g) but not in the nuclear fractions (Supplementary Fig. 8c) of macrophages isolated from *S. japonicum*-infected mice. The similar result showing the interaction between SR-A and IRF5 in

the cytoplasm was also observed in SEA-stimulated macrophages (Supplementary Fig. 8d–e). In addition, confocal results further showed that SR-A and IRF5 co-localized with the Golgi in macrophages (Supplementary Fig. 8f).

On the basis of the above data, we hypothesized that through interaction with IRF5, SR-A may prevent IRF5 nuclear translocation. Indeed, knockdown of SR-A in RAW264.7 cells led to decreased interaction of SR-A and IRF5 in the cytoplasm (Supplementary Fig. 8g). More importantly, IRF5 in the nuclei was significantly increased on SR-A knockdown (Supplementary Fig. 8h,8i). On the contrary, forced expression of SR-A increased the interaction of SR-A and IRF5 in the cytoplasm (Supplementary Fig. 8j) but significantly reduced IRF5 in the nuclei (Supplementary Fig. 8k,l). In addition, although both antigens lead to an endocytosis of membrane SR-A, results showed that antigen from *S. japonicum*, but not *M. tuberculosis*, resulted in the significant increase of the cytoplasmic SR-A in macrophages (Supplementary Fig. 10). Together, these results demonstrated that pathogen modulates SR-A to interact with IRF5 in macrophage cytoplasm, which inhibits IRF5 nuclear translocation.

**Discussion**
Although it has been well recognized that SR-A has an important function in non-infectious diseases[3,38], the role of SR-A in regulating the crosstalk between innate and adaptive immunity to pathogen infection has remained largely unclear. In this study, we for the first time report a SR-A interacts with IRF5 to inhibit its nuclear translocation, resulting in an M2 polarization of macrophages and subsequently favouring adaptive Th2 responses on pathogen infection/stimulation.

Schistosomiasis is one of the most common parasitic infections of humans that affect more than 230 million people worldwide. The most serious immunopathogenesis in schistosomiasis masoni and japonica is the egg-induced granulomatous inflammatory response and fibrosis in the host liver and intestines, which impacts on the hosts' living quality, health status or even mortality[29]. In this study, we show that although SR-A deficiency has little impact on the susceptibility to schistosome infection, it results in reduced granuloma size and fibrosis in the liver and intestine. The granuloma acts to protect the surrounding host tissue or even host life from the toxins released by the eggs by providing a physical barrier between the egg and the tissue, and by sequestering the antigenic products secreted by the egg[39]. In addition, in our study, in a specific stronger M1 and Th1 background in *S. japonicum*-infected SR-A-deficient mice, the relative mild granuloma response may facilitate the transit of eggs through the wall of the intestine and lead to serious breaks in the epithelial barrier, which promotes increased bacteria and toxins

**Figure 4 | SR-A suppresses M1/Th1 but induced M2/Th2 polarization.** (**a**) Two weeks after adoptive transfer of infected mice derived peritoneal macrophages to normal WT mice, Th1 or Th2 cells in recipient mouse splenocytes were analysed. Data are expressed as the mean ± s.d. of 12 mice from two independent experiments, ***$P < 0.001$ (ANOVA/LSD). (**b**) CD4$^+$ T cells from control WT mice were purified, and then co-cultured with purified peritoneal macrophages from infected WT or SR-A-deficient mice. Percentages of Th1 or Th2 cells after co-culture for 48 h were analysed by FACS. (**c**) CD4$^+$ T cells from control WT mice were co-cultured with purified peritoneal macrophages and SEA, and then analysis of Th1 and Th2 cells. (**d–g**) Four weeks after adoptive transfer of normal WT or SR-A-deficient mice derived peritoneal macrophages to infected WT or SR-A-deficient mice. (**d,e**) Paraffin-embedded liver sections stained with H&E. Scale bars, 100 μm. For each mouse, the sizes of 30 liver granulomas around single eggs were quantified with AxioVision Rel 4.7. The statistical results of Th1 and Th2 cells by FACs in the liver (**f**) or spleen (**g**) of recipient mouse were shown. Data are expressed as the mean ± s.d. of six mice for each group, and the experiments were repeated twice with similar results. *$P < 0.05$, **$P < 0.01$, ***$P < 0.001$ (ANOVA/LSD). (**h**) FACS analysis of Th1 and Th2 cells by BMDMs from control WT mice transfected with siRNA targeting SR-A (si-SR-A) or nontargeting siRNA (ctrl-siRNA) and co-culture with WT CD4$^+$ T cells and SEA. (**i**) FACS analysis of Th1 and Th2 cells by BMDMs from control WT mice transfected with pcDNA3.1-MSR (SR-A) or empty vector (pcDNA3.1) and co-culture with WT CD4$^+$ T cells and SEA. (**j**) IL-12p35 or IL-4 mRNA in peritoneal macrophages with or without SEA stimulation. IL-12 or IL-4 expression after BMDMs transfected with si-SR-A or ctrl-siRNA and SEA stimulation by RT-PCR (**k**) or ELISA (**l**). RT–PCR (**m**) or ELISA (**n**) analysis of IL-12 or IL-4 in BMDMs transfected with pcDNA3.1-MSR (SR-A) or empty vector (pcDNA3.1) and SEA stimulation, data are expressed as the mean ± s.d. from three independent experiments, ***$P < 0.001$, **$P < 0.01$, *$P < 0.01$ (ANOVA/LSD).

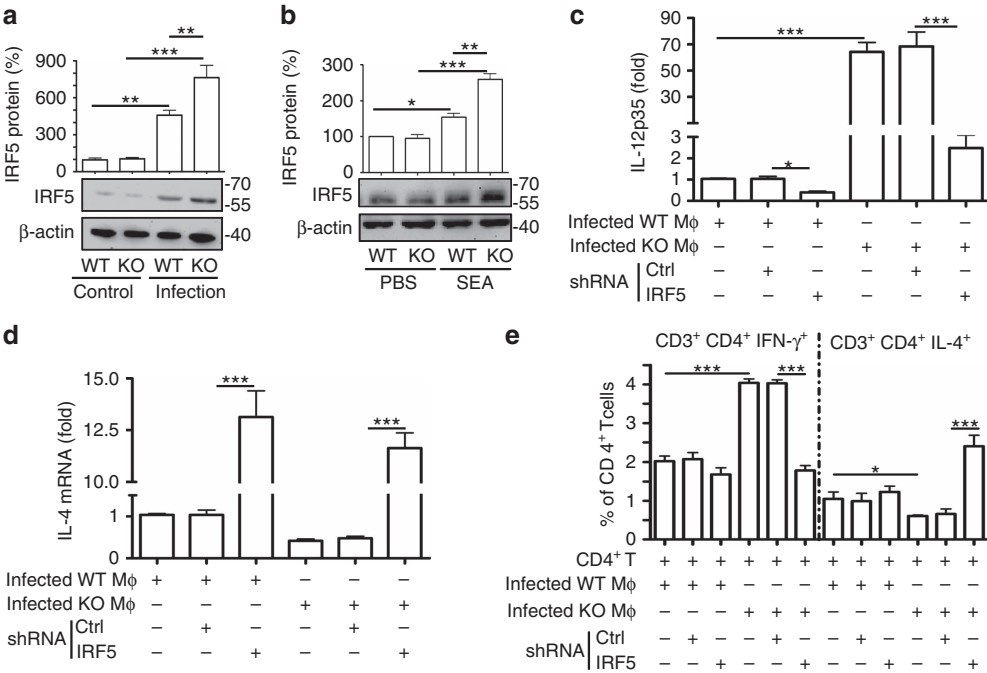

**Figure 5 | SR-A-promoted M2 polarization is associated with decreased IRF5 amounts.** (**a**) Immunoblot analysis of IRF5 in total protein extracts from peritoneal macrophages from WT or SR-A-deficient mice infected with or without *S. japonicum*. Data are expressed as the mean ± s.d. from each group, and are representative of one typical experiment out of three, ***$P < 0.001$, **$P < 0.01$ (ANOVA/LSD). (**b**) IRF5 protein levels in peritoneal macrophages from WT or SR-A-deficient mice after 48 h of stimulation with SEA. (**c-e**) Peritoneal macrophages from *S. japonicum*-infected WT or SR-A-deficient mice were transfected with sh-IRF5 or ctrl-shRNA in the presence of SEA. RT–PCR analysed the mRNA level of IL-12p35 (**c**) or IL-4 (**d**). (**e**) After transfection, peritoneal macrophages were co-cultured with CD4$^+$ T cells from control WT mice stimulated with SEA, and Th1 and Th2 cells were analysed by FACS. Data are expressed as the mean ± s.d. of each group, and are representative of one typical experiment out of three, ***$P < 0.001$, **$P < 0.01$, *$P < 0.05$ (ANOVA/LSD).

from the lumen penetrating the breach, and resulted in the development of lethal endotoxemia[30,40].

As one of the major participants, the CD4$^+$ T cells orchestrate the development of immunopathology in schistosomiasis[29]. Th1 and Treg responses contribute to the downregulation of immunopathology[22,28], while Th2, Th17 and Tfh responses contribute to the upregulation of immunopathology[23,26,27]. In *S. japonicum*-infected mouse model, we show that SR-A deficiency primarily shifts an Th2-type response towards a predominantly Th1 response, which contributes to a significantly decreased immunopathology[30]. Consistently, the loss of SR-A further strongly enhances adaptive Th1 response when injects mice with heat-killed *M. tuberculosis*, a well known pathogen of inducing Th1 responses[41]. We demonstrates a novel function for innate SR-A in the regulation of Th1/Th2 balance in adaptive immunity, suggesting therapeutic potential for targeting SR-A pathway in schistosomiasis.

Macrophages, one of the major cellular constituents of liver granulomas[29] and important immune regulators in schistosomiasis[30,42], are known to retain considerable plasticity and can be categorized into M1 and M2 phenotype. During schistosome infection, M1 macrophages are thought to be cytotoxic to schistosomula through production of nitric oxide (NO), which is an effector mechanism previously implicated in the killing of schistosomula *in vitro*[43], as well as can restrain hepatic fibrosis *in vivo*[31]. In contrast, M2 macrophages are important regulators of fibrosis via Arg-1 activation[44] and CCL17/22 production, which are thought to contribute to the granulomatous and fibrotic development in the liver during schistosomiasis[30,31]. Here for the first time, we show with a *S. japonicum*-infected mouse model that SR-A deficiency results in an impaired M2 but a stronger M1 phenotype in macrophages. Similarly, SR-A deficiency also

strengthens M1 polarization in heat-killed *M. tuberculosis*-treated models. Take together, these findings suggest that SR-A induces M2 but inhibits M1 polarization of macrophages in hosts encountering pathogen infections.

In addition to participation to innate immunity, macrophages also serve as professional antigen-presenting cells (APCs) and play a key role in regulating adaptive immune responses[45]. Thus, phenotypic plasticity of macrophages may regulate adaptive immune responses. It has been reported that M1 macrophages *in vitro* induced from human monocytes efficiently produce proinflammatory cytokines such as IL-12/IL-23 (ref. 46), which are important in promoting Th1/17 responses. In addition, study has shown that in an OVA-specific TCR-transgenic mice model, antigen complexes (IgG-OVA)-activated M2 macrophages can preferentially induce CD4$^+$ T cells to produce higher levels of IL-4 (ref. 47), which is important in promoting Th2 responses. Our *in vivo* and *in vitro* results show that macrophages isolated from *S. japonicum*-infected WT mice or stimulated by SEA show an M2-like phenotype and producing IL-4 (refs 48,49), and preferentially induced Th2-type immune responses. However, SR-A-deficient macrophages isolated from *S. japonicum*-infected SR-A-deficient mice or stimulated by SEA are primary M1 phenotype with IL-12 production, and efficiently promote CD4$^+$ T-cell differentiation towards Th1 phenotype. SR-A has been shown to be also expressed by vascular smooth muscle cells, endothelial cells, human lung epithelial cells, microglia and so on[50]; however, SR-A is predominantly expressed in macrophages, which plays a much more important role than other cells mentioned above in schistosomiasis[29,30,42]. Therefore, we use adoptive transfer experiment to better investigate the role of macrophage in the regulation of CD4$^+$ T-cell differentiation and granuloma formation. Previous studies have shown that dendritic

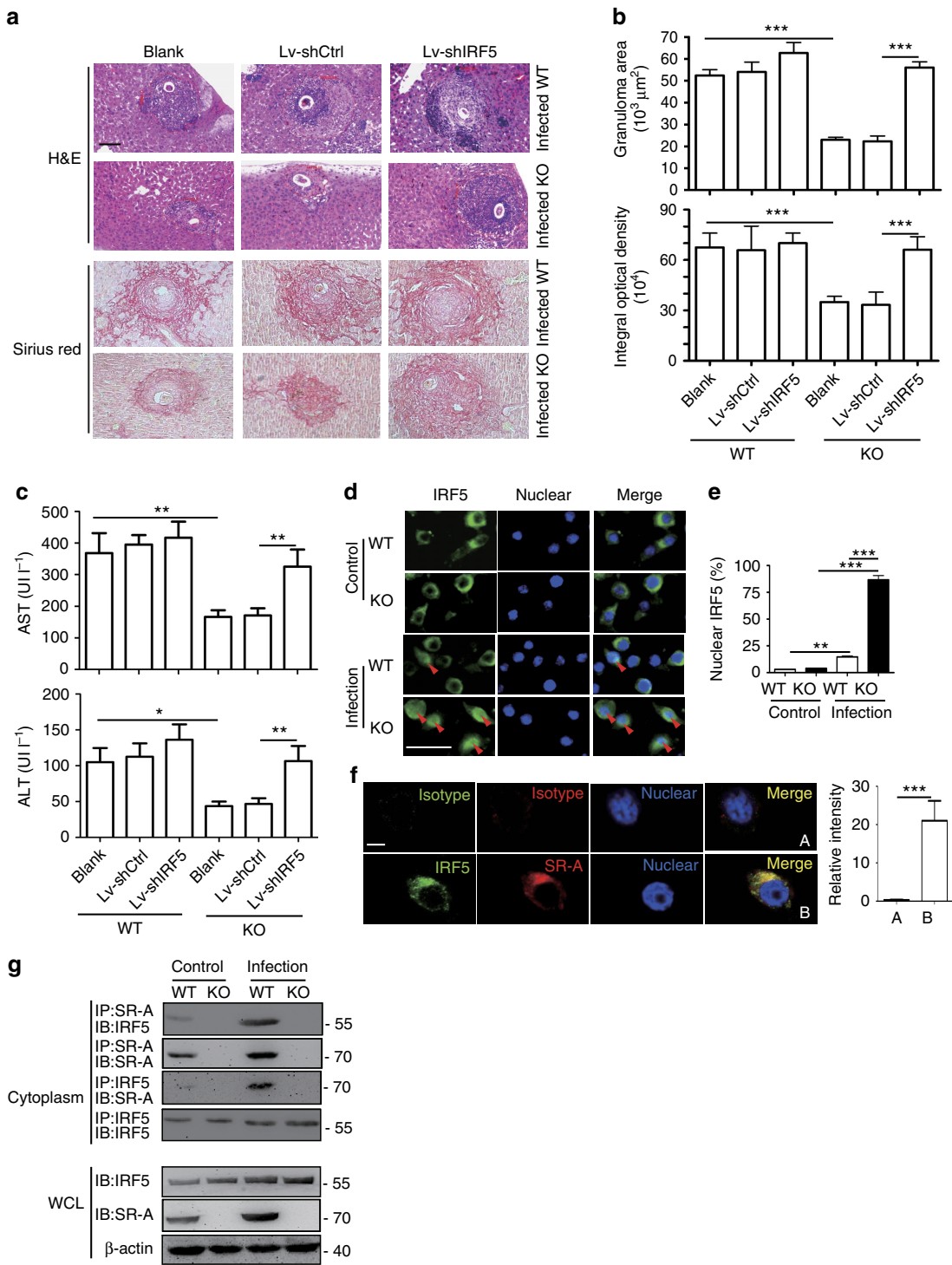

**Figure 6 | SR-A/IRF5 interaction promotes liver pathology by regulation of M2/Th2 polarization. (a–c)** Mice were i.v. injected with shRNA lentiviral particles targeting IRF5 since 3.5 weeks post *S. japonicum* infection by weekly for 4 weeks and killed at 7.5 weeks post *S. japonicum* infection for further study. (**a,b**) Granulomas and fibrosis in mouse livers from *S. japonicum*-infected WT or SR-A-deficient mice after Lv-shRNA injection. Scale bars, 100 μm. (**c**) Levels of serum ALT/AST were determined from *S. japonicum*-infected WT or SR-A-deficient mice after Lv-shRNA injection. Data are representative of two experiments with six mice per group in each experiment. ***$P < 0.001$, **$P < 0.01$, *$P < 0.05$ (ANOVA/LSD). (**d**) Immunofluorescence of IRF5 (green) in the peritoneal macrophages from WT or SR-A-deficient mice infected with or without *S. japonicum*. Scale bars, 50 μm. (**e**) Quantification of nuclear IRF5 in the image above (from at least three fields of view with at least 100 cells per sample). (**f**) Co-localization of endogenous SR-A and IRF5 in RAW264.7 cells by confocal microscopy. Twenty macrophages were randomly selected for cell quantification of SR-A/IRF5 co-localization. For isotype control, 10 macrophages were randomly selected. Scale bars, 10 μm. Relative intensity = intensity of the object (cell) − intensity of the background, quantification of fluorescence in single macrophages of SR-A and IRF5 was achieved by ImageJ software. ***$P < 0.001$. Mean ± s.d. (**g**) Immunoblot analysis of IRF5 of endogenous SR-A immunoprecipitated from cytoplasmic protein extracted from peritoneal macrophages of WT or SR-A-deficient mice infected with or without *S. japonicum*. The association of SR-A and IRF5 was confirmed by a reciprocal immunoprecipitation assay using anti-IRF5. Immunoblot was also carried out using whole cell lysate (WCL).

cells are essential for the induction of $CD4^+$-type immune responses following schistosome infection[51]. However, our study suggests that SR-A likely plays a more important role in macrophages than in DC based on following findings: (1) macrophages, but not DC, expressed much more SR-A after schistosome infection; (2) the deficiency of SR-A in DC only results in a slight and statistically insignificant alteration in Th1/Th2 induction; (3) similar levels of IRF5 are detected in purified DC from SR-A-deficient and WT mice. Thus, our results for the first time indicate that innate SR-A regulates macrophage polarization, which subsequently directs adaptive $CD4^+$ T-cell differentiation.

The transcription factor-IRF5, which is crucial for the activation of inflammatory cytokines including TNF, IL-6, and IL-12 (ref. 37), and the polarization of macrophages[18,35]. The ectopically expressed IRF5 has been shown to translocate to the cell nuclei in response to LPS, which promotes gene induction of proinflammatory cytokines[37]. The binding of IRF5 to its target-gene promoters is necessary for the binding of RNA polymerase II and subsequent transcriptional activation[37]. In this study, we show that the levels of nuclei IRF5 (but not IRF4) and inflammatory cytokines expression are significantly increased in SR-A-deficient macrophages either from mice with *S. japonicum* infection or after *in vitro* stimulation with SEA, suggesting that SR-A inhibits IRF5 function. We speculate that SR-A inhibits the expression of M1-promoted Th1 response mainly through preventing the recruitment of IRF5 to its target-gene promoters.

Study has shown that cell surface SR-A proteins are activated on ligand binding and enter the cell via coated pits through the endocytic pathway[2]. SR-A preferred interacts/transfers lipoid or lipoid-containing molecules[5], which are rich in helminthic egg antigens (for example, SEA) or bacterial (for example, *M. tuberculosis*) lysate. Consistently, we also show that the level of membrane SR-A is decreased after *S. japonicum* infection, or *in vitro* stimulation with SEA or heat-killed *M. tuberculosis*. However, by some unknown mechanisms, stimulation with *S. japonicum* egg antigens, but not *M. tuberculosis*, results in the significant increase in the level of SR-A in macrophage cytoplasm. We speculate that the increased level of cytoplasmic SR-A in macrophages contributes to the regulation of IRF5-mediated macrophage polarization, although SEA (which contains glyco, protein, lipid, glycoprotein, glyco-lipid, lipoprotein and so on) may also bind to many other molecules except for SR-A or even directly enter the cells to alter a cascade response. However, the mechanism behind the regulation of pathogens to SR-A expression needs to be extensively studied in the future.

Subsequently, we show a novel cytoplasmic association between SR-A and IRF5, a key molecule controlling M1 polarization. Many other cytolpasmic molecules, such as TRAF6 or polyethylene glycol cholesteryl ester (PEG-Chol), also interact with SR-A in the cytoplasm[52,53], which supports the possible interaction in our study. The intracellular organelles in which SR-A and IRF5 interact in the cytoplasm remain to be determined. We further show that the binding of SR-A to cytoplasmic IRF5 associates with decreased IRF5 nuclear translocalization. We speculate that the reduced level of nuclear IRF5 significantly decreases M1 but promotes a M2 polarization in macrophages and shifts IL-12/IL-4 production, and then favours development of Th2 but suppresses Th1 induction (Supplementary Fig. 11). Krausgruber *et al.*[35] have shown the importance of IRF5 in the regulation of macrophage polarization; however, the precise mechanism is unknown. Our study further sheds light on significance of SR-A/IRF5 interaction to regulate macrophage polarization and Th1/Th2 differentiation.

In summary, our data have established a new paradigm for pattern recognition receptor regulation of adaptive immunity and suggest the SR-A–IRF5 regulatory axis as a new target for therapeutic intervention to pathogen infection.

## Methods

**Mice.** SR-A-deficient ($Msr1^{-/-}$) mice on ICR (Institute of Cancer Research) background were a kind gift from Prof Chen Qi (Nanjing Medical University), age (6 weeks old) and gender (female)-matched WT mice with identical genetic backgrounds were used as controls. Mice were maintained in the Animal Laboratory Resource Facility at Nanjing Medical University. All experiments were performed in strict accordance with the Regulations for the Administration of Affairs Concerning Experimental Animals (1988.11.1). All animal procedures were approved by the Institutional Animal Care and Use Committee (IACUC) of Nanjing Medical University for the use of laboratory animals.

**Pathogen infection or stimulation and antigen preparation.** *S. japonicum* cercariae were maintained in *Oncomelania hupensis* snails as the intermediate host, which were purchased from Jiangsu Institute of Parasitic Disease (Wuxi, China). Each mouse was infected with $12 \pm 1$ cercariae of *S. japonicum* by abdominal skin exposure. In each independent experiment, 12 mice were randomly chosen from the infected and normal control groups at 8 weeks post infection, and killed for further study.

For other pathogen challenge experiments, each mouse was challenged by i.p. injection of heat-inactivated *M. tuberculosis* (Sigma-Aldrich, St Louis MO, $4 \text{ mg kg}^{-1}$). Splenocytes and peritoneal macrophages were prepared at 24 h after injection for Th1/Th2 and M1/M2 analysis.

SEA was obtained from purified and homogenized *S. japonicum* eggs. The protein concentration of SEA was determined using a bicinchoninic acid Protein Assay kit (Bio-Rad, Richmond, CA).

**Pathology examination.** *S. japonicum*-infected mice were killed and the livers or intestines (ileum) were fixed in 10% neutralized formaldehyde and embedded in paraffin. Multiple liver sections (5 μm thick) in each mouse were dewaxed and stained with haematoxylin and eosin (H&E) for granuloma analysis, or stained with Sirius Red (Sigma-Aldrich), Masson trichrome (Sigma-Aldrich) or α-SMA (Sigma-Aldrich) for fibrosis analysis. For each mouse, the sizes of 30 liver granulomas or 10 intestinal granulomas around single eggs were quantified using AxioVision Rel 4.7 (Carl Zeiss GmbH, Jena, Germany). Data are expressed in area units. Granuloma sizes ($100 \times$) are expressed as means of areas measured in $\mu m^2 \pm$ s.d., and granulomas were analysed using Axiovision software (Carl Zeiss). In addition, fibrosis was determined histologically by measuring the intensity of fibrosis in six random ($100 \times$) digital images captured from collagen-specific sirius red, masson trichrome or α-SMA-stained slides of each mouse using Image-Pro-Plus software 6.0 (Media Cybernetics, Silver Spring, MD, USA). The mean optical density of collagen was determined by dividing integral optical density by the image area.

To determine hepatocyte damage, serum levels of AST and ALT were assayed by Olympus AU2700 Chemical Analyzer (Olympus, Tokyo, Japan).

**Purification of peritoneal and hepatic macrophages.** Peritoneal macrophages were prepared as described previously with some modifications[54]. Briefly, peritoneal exudate cells from each mouse were collected using lavage with ice-cold PBS with 1% FBS. Peritoneal exudate cells were obtained by centrifugation at $500g$ for 5 min at 4 °C and resuspended in RPMI medium containing 10% FBS and 1% penicillin streptomycin. Nonadherent cells were removed after cold PBS wash. Adherent macrophages were incubated with 5 mM EDTA/PBS for 10 min at 37 °C, and then detached by vigorous pipetting to prepare single-cell suspension for purity analysis. Peritoneal adherent cells constituted >98% of macrophages according to FACS analysis ($F4/80^+ CD11b^+$).

Hepatic macrophages were prepared as described previously with some modifications[55]. The liver suspension was prepared as described before and then filtered through a 200-gauge mesh. Hepatocytes were removed by centrifugation at $50g$ for 5 min. Supernatant containing liver mononuclear cells (MNCs) was isolated by density gradient centrifugation with Percoll (25% v/v over 50% v/v) at $800g$ for 30 min to remove eosinophilic granulocytes and other F4/80-positive cells. The layer between the 25 and 50% gradient interface was collected and then were incubated in RPMI medium, the non-adherent cells were removed 3 h later by extensive washing with medium. Hepatic macrophages were identified by using F4/80 and CD11b, two commonly used murine macrophage markers.

**$CD4^+$ T-cell preparation and *in vitro* stimulation.** Briefly, single-cell suspensions of splenocytes and lymphocytes were prepared by mincing the mouse spleens and mesenteric lymph nodes in PBS containing 1% FBS (Gibco, Grand Island, NY) and 1% EDTA. Red blood cells were then lysed using ACK lysis buffer. For preparation of hepatic lymphocytes, mouse livers were perfused via the portal vein with D-Hank's solution (Invitrogen Gibco, Carlsbad, CA, USA). The excised liver

was cut into small pieces and incubated in digestion buffer (collagenase IV/dispase mix, Invitrogen Life Technologies, Carlsbad, CA). The digested liver tissue was then homogenized using a MediMachine with 50 mm Medicons (Becton Dickinson, San Jose, CA) for 5 min at low speed. The liver suspension was then centrifuged at low speed to sediment the hepatocytes. The remaining cells were separated on a 35% Percoll (Sigma-Aldrich) gradient by centrifuging at 600g. The lymphocyte fraction was resuspended in red cell lysis buffer and then washed in 10 ml of complete RPMI1640 with 1% EDTA.

Splenocytes from WT or SR-A-deficient mice were stimulated with or without SEA (50 µg ml$^{-1}$) for 48 h, and the proportions of CD4$^+$ T-cell subsets: CD3$^+$CD4$^+$IFN-γ$^+$ (Th1), CD3$^+$CD4$^+$IL-4$^+$ (Th2), CD3$^+$CD4$^+$IL-17$^+$ (Th17), CD3$^+$CD4$^+$CXCR5$^+$PD-1$^+$ (Tfh) and CD4$^+$CD25$^+$Foxp3$^+$ (Treg) were analysed by FACS.

Macrophages from WT or SR-A-deficient mice were stimulated with or without SEA (50 µg ml$^{-1}$) for 48 h, and the M1 or M2-related molecules were analysed by FACS or RT–PCR.

CD4$^+$ T cells were purified by negative selection using a CD4$^+$ T-Cell Isolation kit (lot:5140218259; Miltenyi Biotec, Bergisch Gladbach, Germany) and a magnetic-activated cell sorter according to the manufacturer's recommendations. Cells were co-cultured with macrophages in the presence of SEA (50 µg ml$^{-1}$) for 48 h. Then, CD4$^+$ T cells were examined using surface staining (see R10 in Supplementary Fig. 12a).

**Flow cytometry.** The following antibodies were used for flow cytometry: F4/80-FITC (rat-anti-mouse; BM8; used at 1:200 dilution), CD11b-APC (rat-anti-mouse; M1/70; used at 1:200 dilution), CD3e-APC (Armenian Hamster-anti-mouse; 145-2C11; used at 1:200 dilution), CD4-FITC (rat-anti-mouse; RM4-5; used at 1:200 dilution), CD25-APC (rat-anti-mouse; PC61.5; used at 1:300 dilution), Foxp3-PE (rat-anti-mouse; FJK-16s; used at 1:40 dilution), IFN-γ-PE (rat-anti-mouse; 11B11; used at 1:40 dilution), IL-4-PE (rat-anti-mouse; 11B11; used at 1:40 dilution), IL-17A-PE (rat-anti-mouse; eBio17B7; used at 1:40 dilution), CD3e-Percp-cy5.5 (Armenian Hamster-anti-mouse; 145-2C11; used at 1:200 dilution), PD-1-PE (rat-anti-mouse; RMP1-30; used at 1:40 dilution), CD16/32-PE (rat-anti-mouse; 93; used at 1:400 dilution) (all from eBioscience, San Diego, CA), CD206-PE (goat-anti-mouse; FAB2535P; used at 1:10 dilution) and CXCR5-APC (rat-anti-mouse; 2G8; used at 1:50 dilution; BD Pharmingen, San Diego, CA).

For Th1/Th2/Th17 analysis, $2 \times 10^6$ of single-cell suspension were cultured in complete RPMI 1640 medium (Gibco, Grand Island, NY) and stimulated with 25 ng ml$^{-1}$ PMA (cat: P1585; Sigma) and 1 µg ml$^{-1}$ ionomycin (cat: I0634; Sigma) in the presence of 0.66 µl ml$^{-1}$ Golgistop (cat:554724; BD, San Jose, CA) at 37 °C in 5% CO$_2$ for 6 h. Cells were stained for surface molecules (CD3-APC and CD4-FITC), fixed and permeabilized with Cytofix/Cytoperm buffer and then intracellularly stained with PE-conjugated antibodies against IFN-γ, IL-4, IL-17 or isotype control antibody, respectively.

For Treg analysis, $2 \times 10^6$ of single cell suspensions were surface stained with CD4-FITC and CD25-APC. Then, cells were fixed and made permeable with fixation-permeabilization buffers (eBioscience) and blocked with Fc receptor (rat-anti-mouse; 93; used at 1:200 dilution; eBioscience, San Diego, CA). Finally, cells were stained with PE-conjugated anti-Foxp3 antibodies (cat: 00-5523; eBioscience).

For Tfh analysis, $2 \times 10^6$ cells were surface stained with CD3e-Percp-cy5.5, CD4-FITC, CXCR5-APC and PD-1-PE.

The viability of purified macrophages was assessed by staining with F4/80-FITC and CD11b-APC. For M1/M2 macrophage analysis, cells were preincubated with mouse Fc Block for 15 min to block the nonspecific binding of antibodies, and then stained with antibodies of CD16/32-PE or CD206-PE antibodies.

All cells were subsequently detected using a FACSCalibur flow cytometer (BD Biosciences, Heidelberg, Germany) and analysed by CellQuest software (BD Biosciences) or FlowJo software (Treestar, Inc., San Carlos, CA).

**Quantitative RT–PCR.** RNA was extracted with RNeasy Mini Kit (cat: 74104; Qiagen, Hilden, Germany). Total RNA quantity was measured using a BioPhotometer (Eppendorf, Hamburg, Germany). Reverse transcription from 2 µg of RNA was performed using the RevertAid First strand cDNA Synthesis Kit (cat: K1621; Fermantas Life Sciences, St. Leon-Rot, Baden-Württemberg, Germany). Real-time PCR was performed using Power Syber Green PCR Master Mix (cat: 4309155; Applied Biosystems, Foster City, CA) and detected by the 7300HT Fast Real-Time System (Applied Biosystems). The sequences of gene-specific primers were published previously[56] (Supplementary Table 1). Data were processed using SDS software (Applied Biosystems). Results were normalized to the expression of the housekeeping gene GAPDH.

The heat maps of mRNA abundance were generated using the Hemi IBP software (Heatmap Illustrator, version 1.0.3.3). Results are presented as higher (red) or lower (blue) expression after infection or stimulation (key (below), log$_2$ fold change).

The cycling parameters were as follows: stage 1, 50 °C for 2 min; stage 2, 95 °C for 10 min, stage 3, 40 cycles of 95 °C for 15 s and 60 °C for 1 min, which were concluded by the melting curve analysis process, and fold changes of gene expression were calculated using the $2^{-\Delta\Delta Ct}$ method.

**ELISA and serum analysis.** The homogenized liver tissue was centrifuged at 10,000 r.p.m. for 15 min, and the supernatant was recovered and stored at −80 °C until analyses. Levels of IL-12p70 and IL-4 in serum, liver homogenates or culture supernatant were quantified with ELISA (cat: BMS6004 and cat: BMS163; eBioscience).

Measurement of serum LPS was performed using Toxin Sensor TM Chromogenic LAL Endotoxin Assay Kit (cat: L00350; GenScript, Piscataway, NJ, USA) per the manufacturer's specifications.

The levels of serum AST and ALT were assayed by Olympus AU2700 Chemical Analyzer (Olympus, Tokyo, Japan).

**Adoptive transfer experiments.** Peritoneal macrophages from WT or KO mice with *S. japonicum* infection were isolated as described above, and were then resuspended in PBS and injected i.v. into the tail of control WT mice ($2 \times 10^6$ cells per mouse). Mice were killed 14 days after transfer to investigate the Th1 or Th2 cells by FACS.

Peritoneal macrophages from normal WT or SR-A KO mice were isolated, and were then resuspended in PBS and injected i.v. into the tail of WT or KO mice 3.5 weeks post *S. japonicum* infection, respectively ($2 \times 10^6$ cells per mouse). Mice were killed 4 weeks after transfer to investigate the liver granuloma size and Th1 or Th2 cells in the liver and spleen by FACS.

**Preparation of nuclear and cytosolic extracts.** Nuclear and cytosolic fractions were prepared by using the Nuclear Extraction kit (cat: 2900; Millipore-Upstate, Billerica, MA, USA) according to the manufacturer's instructions. Protein concentrations were determined by using the Bio-Rad Protein Assay kit (Bio-Rad, Hercules, CA, USA).

**Co-immunoprecipitation and western blotting.** Co-immunoprecipitation technique was applied for determination of the interaction between two proteins. Briefly, the lysates (~400 µg of protein) were incubated at 4 °C in rocking condition with anti-SR-A (rat anti-mouse; 2F8; used at 1:1,000 dilution; BMA Biomedical, Basel, Switzerland) or anti-IRF5 (rabbit anti-mouse; cat: 4950; used at 1:1,000 dilution, Cell Signaling, Danvers, MA). After 2 h, the immune-complexes were incubated with protein A/G-plus agarose beads (cat: sc-2003; Santa Cruz Biotechnology) overnight at 4 °C. The immunopurified proteins were extensively washed and immunoblotted using specific antibodies. Input lysates were blotted by anti-β-actin for protein loading.

For western blot analysis, equivalent amount of protein (100–300 µg) in each group was loaded in each lane for separation by SDS–PAGE and were then transferred onto Immu-Blot PVDF membranes (Bio-Rad, Hercules, CA, USA). After blocking in Tris-buffered saline containing Tween-20 (0.1%) (T-TBS) and milk (5%) at room temperature for 2 h, the membrane was incubated at 4 °C overnight with primary antibodies: rat anti-SR-A (BMA Biomedical), rabbit anti-IRF5 (Cell Signaling), rabbit anti-IRF4 (rabbit anti-mouse; cat: 4948; used at 1:1,000 dilution, Cell Signaling), rabbit anti-Histone H3 (rabbit anti-mouse; A3S; used at 1:1,000 dilution, Millipore, Temecula, CA) or rabbit anti-β-actin (13E5; Cell Signaling), respectively. After washing with T-TBS, membranes were incubated at room temperature for 1 h with HRP-conjugated anti-rat IgG (cat: 7077; used at 1:1,000 dilution), or anti-rabbit IgG (cat: 7074; used at 1:1,000 dilution) secondary antibody (Cell Signaling). After washing, the blots were developed using the ECL system (GE Healthcare, Little Chal-font, Buckinghamshire, UK) following the manufacturer's instructions. The density of each band was quantified by densitometric analysis with ImageJ software (Image Processing and Analysis in Java, National Institutes of Health, USA). Full-length blots are located in Supplementary Fig. 13.

**Immunofluorescence and confocal microscopy.** Macrophages ($3 \times 10^5$) were seeded onto coverslips in six-well dishes. Cells were fixed with 4% paraformaldehyde and permeabilized with 0.5% Triton X-100 in PBS for 15 min at room temperature. Unspecific binding sites were blocked with PBS + 2%BSA for 30 min at 4 °C. Cells were stained with rat anti-mouse SR-A (BMA Biomedicals) and rabbit anti-mouse IRF5 (Cell Signaling). After being stained, excess antibody was removed by four wash steps with PBS + 0.2% Triton X-100, and then cells were staining with PE-conjugated goat anti-rat IgG (cat: ab7010; used at 1:200 dilution; Abcam, Cambride, UK) and Alexa Fluor 488-goat anti-rabbit IgG (cat: ab150077; used at 1:200 dilution; Abcam, Cambride, UK). Finally, cells were washed four times with PBS + 0.2% Triton X-100 and then were mounted in Fluoromount-DAPI (cat: 0100-20; SouthernBiotech, Birmingham, USA) and Golgi-Tracker Red (used at 1:100 dilution; Beyotime Institute of Biochemistry, China) at 4 °C for 30 min. Cells were analysed with a microscope (Carl Zeiss) or Zeiss LSM 710 laser-scanning confocal microscope (Carl Zeiss). Quantification of fluorescence in single macrophages of SR-A and IRF5 was achieved by ImageJ software (http://ima-gej.nih.gov/ij/). Relative intensity = intensity of the object (cell) − intensity of the background[57]. Twenty macrophages were randomly selected for cell quantification of SR-A/IRF5 co-localization. For isotype control, 10 macrophages were randomly selected. ***$P < 0.001$. Mean ± s.d.

**Preparation of BMDMs and cell transfection.** BMDMs were obtained and cultured as described previously with some modifications[58]. Briefly, mice were killed and both femurs were dissected and the adherent tissues were removed. The ends of their bones were cut off, and their marrow tissues were flushed via irrigation with PBS. The cells were cultured in RPMI medium containing 10% foetal bovine serum and 20 ng ml$^{-1}$ macrophage-colony stimulating factor (cat: 315-02; 20 ng ml$^{-1}$ PeproTech, London, UK). After 7-day culture, the macrophages were obtained as a homogeneous population of adherent cells and were maintained at 37 °C in a humidified 5% $CO_2$ atmosphere. The analysis of the cell surface markers showed that more than 98% of the cells expressed macrophage-specific markers (F4/80 and CD11b, see R2 in Supplementary Fig. 12c).

Transfection of BMDMs or RAW264.7 cell line was carried out as previously described[59]. Briefly, $5 \times 10^6$ cells were resuspended in Opti-MEMI reduced serum media (GIBCO, NY, USA) and then were transfected with pcDNA3.1-MSR (kind gift from Prof. Chen Qi in Nanjing Medical University), SR-A siRNA pool (GenePharma, Shanghai, China) or Lv-shRNA-IRF5 (GenePharma) by using Lipofectamine 2000 (cat: 11668-019; Invitrogen) according to the manufacturer's protocol. Cells were used 1–3 days after transfection. The Lv-shRNA-IRF5 target sequence was 5′-GCTCTTCTACAGCCAGCTAGA-3′. The Lv-shRNA-NC target sequence was 5′-TTCTCCGAACGTGTCACGT-3′.

**Lentivirus injection.** Lentiviral packaging was carried out as previously described with some modifications[60]. Briefly, HEK-293T cells were transfected with lentiviral packaging vectors (pLV/helper-SL3, pLV/helper-SL4 and pLV/helper-SL5) as well as the shRNA–IRF5 plasmid. The supernatants were collected and vectors concentrated by ultracentrifugation. The LV titres were determined by the method of end point dilution through counting the numbers of infected green cells (based on GFP expression) at ×100 magnification under fluorescence microscope (Olympus, Tokyo, Japan) 96 h after infection to 293T cells. Titre in IU ml$^{-1}$ = (the numbers of green fluorescent cells) × (dilution factor)/(volume of virus solution).

Lentiviral supernatant was diluted in 0.9% saline (Delta-Pharma, Pfullingen, Germany) and polybrene (8 µg ml$^{-1}$ final concentration; Sigma) to give a dose of $5 \times 10^7$ TU in a 100 µl injection volume. Mice were injected with lentiviral vectors (100 µl) into the tail vein since 3.5 weeks post *S. japonicum* infection by weekly for 4 weeks.

**Statistical analysis.** All analyses were carried out with the SPSS 19.0 software. Data were shown as mean ± s.d. The significance of difference between two groups was identified using a Student's *t*-test. Multiple comparisons were performed by one-way ANOVA, and followed by LSD post test for comparison between two groups. Survival between infected WT and KO mice was compared by Kaplan–Meier survival curves with log-rank test. *P* values <0.05 were considered significant. Significant differences were as follows: *$P<0.05$, **$P<0.01$, ***$P<0.001$.

**Data availability.** The authors declare that the data supporting the findings of this study are available within the article and its Supplementary Information files, or from the authors on reasonable request.

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

## Acknowledgements
We thank Dr Chen Qi (Nanjing Medical University) for providing SR-A-deficient mice and pcDNA3.1-MSR1, Dr Antonio Sica (University of Piemonte Orientale A) for providing most M1- and M2-related gene primers. We also thank Hui Bai (Nanjing Medical University) for valuable technical assistance. This work was supported by the grant from the National Natural Science Foundation of China (No. 81430052) to Chuan Su.

## Author contributions
C.S., Z.X., X.C. and Q.C. conceived and designed the experiments. Z.X. and C.S. analysed the data. Z.X., L.X., W.L., X.J., X.S., X.C., J.Z., S.Z., Y.L., W.Z., X.D., X.Y., F.L. and H.B. performed the experiments. Manuscript was written by C.S. and Z.X. All authors read and approved the final manuscript.

## Additional information

**Competing interests:** The authors declare no competing financial interests.

