## [Peer Review File · Nature Communications]

Reviewers' comments:

Reviewer 1 (Remarks to the Author):

In this manuscript, Xu et al. attempt to make a link between SR-A, a scavenger receptor (one of many), and macrophage polarization, mediated by IRF5. Overall, the data and conclusions drawn are preliminary in nature. Many alternative explanations for the data are possible. Being too brief, too wordy, or far too assertive confuses the writing. The experimental design is poor. Overall, this manuscript is not suitable for publication.

Specific comments (not in order of importance):

1. Abstract. Stating SR-A is 'important' and 'critical' is nonsensical writing. Just state the facts.
2. In figure 1, which is hardly discussed in the results section, the authors demonstrate the SR-A-deficient mice (proper genetic nomenclature not used, presumably *Msr1*) have smaller granulomas (that is not explained or quantified properly) and die earlier than controls. These data are questionable. The authors state in the legend that 12 mice were used per group/experiment, and it was done twice. Is the data in 1a the combined data or one experiment? In either case, the 'significance' may be mathematically true, but is biological irrelevant, as both infected groups die between 15-20 weeks. In 1b the authors state they quantified 30 granulomas - is this from 1 mouse, or 30 from all mice? Essentially, and without going in to too much detail, this experimental series was not performed or analyzed properly, and the data cannot be interpreted.
3. Line 3, pg. 3. 'or in the cytoplasm of macrophages'. This is not sufficiently specific.
4. Line 13-20, pg. 3. Entirely unnecessary.
5. pg. 4. The authors exhibit a limited understanding of macrophage polarization. For example, *Arg1* is also highly expressed in M1 macrophages (El Kasmi et al. and many other references) but by a different mechanism.
6. pg. 5. The explanation surrounding the T cell responses are not sufficiently developed. The authors need to measure cytokine production in liver T cells directly isolated from infected tissue.
7. pg. 5, line 22. 'which was reported to elicit a Th1 response' seems unnecessary.
8. The data concerning the heat maps in figure 3 are impossible to interpret. The authors need to test macrophage polarization from macrophages isolated (or stained) from granulomas. Why were peritoneal macrophages used? There are no criteria for 'differences' between genotypes, and it is not clear how reproducible the data was. Why wasn't IL-4 used as a control?
9. pg. 7. A mixed bone marrow chimera is required to understand the SR-A intrinsic versus

extrinsic effects. The siRNA experiments are difficult to interpret, if not impossible.

10. On pg. 8 the authors finally get to the issue of IL-12, which might be responsible for the granuloma phenotype in Figure 1 (e.g. Wynn's data on IL-12 and schistosome granulomas). However, the data here is hard to interpret without enumeration of IL-12 within the granulomas. The experiments shown are indirect and inconclusive.

11. The data concerning IRF5 regulation is too preliminary in nature to interpret. There is no compelling reason to imagine a cell surface scavenger receptor would regulate and archetypal member of the IRF family. A co-crystal structure, or compelling biochemical evidence would be essential.

Reviewer 2 (Remarks to the Author):

The manuscript by Xu et al. is exciting because of novel data identifying a molecular mechanism for macrophage polarization and also because of new information about the role of alternatively activated macrophages in type 2 pathology. The data suggests that M2 polarization is critical for granuloma development and fibrosis in response to *S. japonicum* at the time point shown.

The authors support their conclusions effectively with well-controlled experiments. They show that macrophages from SR-A KO mice do not exhibit normal polarization to an M2 phenotype in the presence of type 2 stimuli. They explain that the immune dysregulation is due to SR-A from macrophages using adoptive transfer and siRNA transfection/overexpression approaches. They also use sh-IRF5 to demonstrate that IRF5 is necessary for the SR-A KO phenotype.

While I find the data convincing, I have a few comments and questions.

1) I think the observations about SR-A during *S. japonicum* infection themselves are interesting to immunology and parasitology communities, but the manuscript would be stronger if the authors added more context to their findings. The field still lacks good tools to study precise roles of M2 polarized macrophages. It would be helpful if the authors explained how prevalent SR-A-expressing macrophages are. Is the KO affecting a subset of M2 macrophages or all of them during *S. japonicum* infection? Do the authors think there are SR-A/IRF5-independent mechanisms of polarization towards an M2 phenotype? Showing this phenomenon is not *S. japonicum*-specific could also help to address this point.

2) To add to the previous point, it would also be useful if the authors included a reference to Vannella et al. PLoS Pathog 10: e1004372 (2014) in the discussion. That publication described that removing IL-4Ralpha signaling on resident macrophages resulted in a failure to downmodulate granuloma size in *S. mansoni* infection. It would be helpful for the authors to speculate about reasons for these different results.

3) Krausgruber et al. Nature Immunology 12, 231-238 (2011) has previously shown the importance of IRF5 in macrophage polarization so it warrants more of a discussion as well to put the new findings into context.

4) The authors should explain why they used CD16/32 as a marker of M1 cells. Did its expression track with MHC class II or CD80/86? It would also be helpful for the authors to include a diagram of flow plots as an example of the macrophage gating strategy.

5) The results/methods/figure legends should be clarified in some spots. For example, it is not clear what time-point is used for the pathology shown in Figure 6. In the Figure 1 legend, the micrographs of intestine tissue are not labeled. These changes will help make the experiments more reproducible and interpretable.

Sincerely,

Kevin Vannella, PhD

We appreciate the helpful and constructive comments of the reviewers. We have carried out the additional experiments suggested by the reviewers and added a few new figures (Figure 3c, Supplementary Fig.3a, Supplementary Fig.4g-4n, Supplementary Fig. 5g). We have also extensively revised our manuscript. The newly acquired results further supported the conclusions described in the manuscript.

Our point-by-point responses to the reviewers' comments are described below:

Reviewers' comments:

Reviewer 1 (Remarks to the Author):

In this manuscript, Xu et al. attempt to make a link between SR-A, a scavenger receptor (one of many), and macrophage polarization, mediated by IRF5. Overall, the data and conclusions drawn are preliminary in nature. Many alternative explanations for the data are possible. Being too brief, too wordy, or far too assertive confuses the writing. The experimental design is poor. Overall, this manuscript is not suitable for publication.

Specific comments (not in order of importance):

1. Abstract. Stating SR-A is 'important' and 'critical' is nonsensical writing. Just state the facts.

Response: We have followed the reviewer's suggestion to delete the “important” and “critical”.

2. In figure 1, which is hardly discussed in the results section, the authors demonstrate the SR-A-deficient mice (proper genetic nomenclature not used, presumably Msr1) have smaller granulomas (that is not explained or quantified properly) and die earlier than controls. These data are questionable. The authors state in the legend that 12 mice were used per group/experiment, and it was done twice. Is the data in 1a the combined data or one experiment? In either case, the 'significance' may be mathematically true, but is biological irrelevant, as both infected groups die between 15-20 weeks. In 1b the authors state they quantified 30 granulomas - is this from 1 mouse, or 30 from all mice? Essentially, and without going in to too much detail, this

experimental series was not performed or analyzed properly, and the data cannot be interpreted.

Response:

We are sorry that we had failed to describe the results in detail nor discuss more deeply. In Results section we had only described the granuloma surround eggs in livers and intestines of *S. japonicum*-infected SR-A^{-/-} mice, but had not mentioned more pathology else in detail. In our study, in addition to smaller granuloma in both livers and intestines, the SR-A^{-/-} mice showed that the integrity of the gut epithelium was compromised, and the epithelial barrier was broken (Fig 1f), which facilitated the development of lethal endotoxemia in SR-A^{-/-} mice (Supplementary Fig. 1d). Consistently, studies^{1,2} have already shown that although the egg granuloma size in intestine is milder, stronger M1/Th1 responses (just like what happened in SR-A^{-/-} mice, which was shown in Fig. 2a-2b and 3a-3c) significantly lead to breaks in the epithelial barrier that causes septicemia as increased numbers of lumen bacteria and/or toxin penetrate the breach. In conclusion, in our study, a decreased egg granuloma in livers and intestines accompany with severer broken epithelial barrier (Fig. 1f), higher endotoxin-mediated toxicity (Supplementary Fig. 1d), stronger M1/Th1 responses (Fig. 2a-2b, 3a-3c), and rapidly succumb to *S. japonicum* infection (Fig 1a) were showed in the SR-A^{-/-} mice. We have modified the Results section (page 6 lines 6-10) and discussed more in our revised manuscript in section of Discussion on page 14 lines 13-22.

Following the reviewer's suggestion, we have added the genetic nomenclature, Msr1, of the SR-A^{-/-} mice in Materials and Methods in our revised manuscript on page 20 line 3.

Experiment in Fig. 1a had been repeated twice with similar results but we had only shown a representative experiment (12 mice per group) in two experiments in our previous manuscript. We have combined the two sets of data in new Figure 1a (24 mice per group) in our revised manuscript.

Given that one worm pair of *S. japonicum* laid approximately 2,200 eggs per day³, it is far (up to six folds) more than *Schistosoma mansoni* worm pairs. In our study, each mouse was infected with 12 cercariae of *S. japonicum*, which is an extremely severe infection for mice, if considering worm number/kg host weight (equal to 24000-30000 worms in a person). Therefore, it is impossible for mice to survive too long (or even for ever) without immediate Praziquantel treatment. In this case, the

extension of mice survival span from 15 to 20 weeks is biologically/medically significant and relevant. The similar situation was also happened in field of cancer treatment research or some other fields.

We have described in detail for the counting of granuloma in the section of Methods in page 21 line 4-7 or in the section of Figures and Legends in page 38 line 8-9 in our revised manuscript: For each mouse, the sizes of 30 liver granulomas or 10 intestinal granulomas around single eggs per mouse were quantified with AxioVision Rel 4.7.

1. Barron L, Wynn TA. Macrophage activation governs schistosomiasis-induced inflammation and fibrosis. *European journal of immunology* **41**, 2509-2514 (2011).
2. Herbert DR, *et al.* Alternative macrophage activation is essential for survival during schistosomiasis and downmodulates T helper 1 responses and immunopathology. *Immunity* **20**, 623-635 (2004).
3. Cheever AW, Macedonia JG, Mosimann JE, Cheever EA. Kinetics of egg production and egg excretion by *Schistosoma mansoni* and *S. japonicum* in mice infected with a single pair of worms. *Am J Trop Med Hyg* **50**, 281-295 (1994).

3. Line 3, pg. 3. 'or in the cytoplasm of macrophages'. This is not sufficiently specific.

Response: Given that SR-A expressed on the Golgi apparatus of macrophages^{1,2}, we have described this sentence more specific as described below. 'or on the Golgi apparatus of macrophages' in our revised manuscript page 3 line3.

1. Sano H, *et al.* The microtubule-binding protein Hook3 interacts with a cytoplasmic domain of scavenger receptor A. *The Journal of biological chemistry* **282**, 7973-7981 (2007).
2. Mori T, *et al.* Endocytic pathway of scavenger receptors via trans-Golgi system in bovine alveolar macrophages. *Laboratory investigation; a journal of technical methods and pathology* **71**, 409-416 (1994).

4. Line13-20, pg. 3. Entirely unnecessary.

Response: We have modified these sentences in our revised manuscript to better fit our story and significance.

5. pg. 4. The authors exhibit a limited understanding of macrophage polarization. For example, Arg1 is also highly expressed in M1 macrophages (El Kasmi et al. and many other references) but by a different mechanism.

Response: The expression of Arg1 could also be upregulated in M1 macrophages under some specific circumstances, e.g. combination of activation of TLR-Myd88 signaling by some specific intracellular pathogens^{1, 2}. While it is widely accepted that M2 macrophages induced under regular conditions, including IL-4 prime or helminth infection, are characterized by high expression of Arg-1^{3, 4, 5, 6, 7}. More important, macrophage classification usually requires the integration of a multiparameter of the markers including arginase, chemokines, costimulation molecules, and cytokines etc, which are included in our study.

1. El Kasmi KC, *et al.* Toll-like receptor-induced arginase 1 in macrophages thwarts effective immunity against intracellular pathogens. *Nature immunology* **9**, 1399-1406 (2008).
2. Qualls JE, *et al.* Arginine usage in mycobacteria-infected macrophages depends on autocrine-paracrine cytokine signaling. *Sci Signal* **3**, ra62 (2010).
3. Porta C, *et al.* Tolerance and M2 (alternative) macrophage polarization are related processes orchestrated by p50 nuclear factor kappaB. *Proceedings of the National Academy of Sciences of the United States of America* **106**, 14978-14983 (2009).
4. Satoh T, *et al.* The Jmjd3-Irf4 axis regulates M2 macrophage polarization and host responses against helminth infection. *Nature immunology* **11**, 936-944 (2010).
5. Cao Q, *et al.* IL-25 induces M2 macrophages and reduces renal injury in proteinuric kidney disease. *Journal of the American Society of Nephrology : JASN* **22**, 1229-1239 (2011).

6. Besnard AG, *et al.* IL-33-mediated protection against experimental cerebral malaria is linked to induction of type 2 innate lymphoid cells, M2 macrophages and regulatory T cells. *PLoS pathogens* **11**, e1004607 (2015).
7. Kambara K, *et al.* In vivo depletion of CD206+ M2 macrophages exaggerates lung injury in endotoxemic mice. *The American journal of pathology* **185**, 162-171 (2015).

6. pg. 5. The explanation surrounding the T cell responses are not sufficiently developed. The authors need to measure cytokine production in liver T cells directly isolated from infected tissue.

Response: Our results had already shown that the expression of IFN- γ in liver CD4⁺T cells (as shown in Fig. 2a) was increased in *S. japonicum* infected SR-A^{-/-} mice, while the expression of IL-4 in liver CD4⁺T cells was decreased (as shown in Fig. 2b).

7. pg. 5, line 22. 'which was reported to elicit a Th1 response' seems unnecessary.

Response: As suggested by the reviewer, we have removed this sentence.

8. The data concerning the heat maps in figure 3 are impossible to interpret. The authors need to test macrophage polarization from macrophages isolated (or stained) from granulomas. Why were peritoneal macrophages used? There are no criteria for 'differences' between genotypes, and it is not clear how reproducible the data was. Why wasn't IL-4 used as a control?

Response: Although it is technically much more difficult for us to isolate and obtain sufficient number of live macrophages with high-purity, especially from uninfected control mice, both mice liver and peritoneal CD16/32-macrophages and CD206-macrophages had been FACS analyzed and similar results had been obtained. As suggested by the reviewer, we have replaced with the RT-PCR results with liver macrophages (Figure 3c), which showed that the expression of M1 related genes was

increased but M2 related genes were decreased in liver macrophages of SR-A^{-/-} mice compared to those of WT mice after *S. japonicum* infection.

We had detected the level of IL-4 but we had put it in Fig. 4f of previous manuscript as a separate figure. Following the reviewer's suggestion, we have added the control data of IL-4 both in the Figure 3c and 3d of our revised manuscript.

9. pg. 7. A mixed bone marrow chimera is required to understand the SR-A intrinsic versus extrinsic effects. The siRNA experiments are difficult to interpret, if not impossible.

Response: We are sorry that we are unable to carry out the mice experiment with mixed bone marrow chimera under the limitation of our experimental conditions. As a replacement, we have carried out additional adoptive transfer experiments to understand the SR-A intrinsic versus extrinsic effects. Results in Supplementary Fig.4g-n in our revised manuscript showed that transferring of SR-A^{-/-} macrophages to *S. japonicum*-infected WT mice dampened the granuloma size and Th2 response, but increased Th1 response both in spleen and liver, indicated a SR-A intrinsic role for macrophages in the development of liver granuloma. However, transferring of WT macrophages to *S. japonicum*-infected SR-A^{-/-} mice increased the liver granuloma size and Th2 response, but decreased Th1 response. We have added these results in our revised manuscript in page 9 lines 7-15.

10. On pg. 8 the authors finally get to the issue of IL-12, which might be responsible for the granuloma phenotype in Figure 1 (e.g. Wynn's data on IL-12 and schistosome granulomas). However, the data here is hard to interpret without enumeration of IL-12 within the granulomas. The experiments shown are indirect and inconclusive.

Response: As suggested by the reviewer, we have detected the expression of IL-12 in liver homogenates by ELISA. Results showed that the expression of IL-12 in the liver of SR-A^{-/-} mice was higher than that in WT mice. We have added the result in Supplementary Fig. 5g

11. The data concerning IRF5 regulation is too preliminary in nature to interpret. There is no compelling reason to imagine a cell surface scavenger receptor would regulate and archetypal member of the IRF family. A co-crystal structure, or compelling biochemical evidence would be essential.

Response: Studies have shown that SR-A not only expressed on the membrane of macrophages, but also in the cytoplasm of macrophages because: 1) the cell surface SR-A proteins are activated upon ligand binding and internalized into cytoplasm via coated pits and traffic the endocytic pathway, 2) recycling between early endosomal compartments and the cell surface [1, 2, 3]. Consistent with our result, studies have shown that many other cytolpasmic molecules, such as TRAF6 or polyethylene glycol cholesteryl ester (PEG-Chol), also interact with SRA in the cytoplasm [3, 4].We have added these in the section of Discussion in page 18 line 5-7. In addition, in our study, SR-A was detected both on the membrane (by FACS) (Supplementary Fig.8g, 8i, and 8n) and in the cytoplasm (by confocal) (Figure7e, Supplementary Fig.8c and 8j), and a SR-A-IRF5 interaction by confocal within the cytoplasm of macrophages in Figure7e and Supplementary Fig.8g.

It is a pity that we are not able to complete the co-crystalstructure analysis by our current technology and condition. Instead, we used STRINE database (http://string.embl.de/) to predict the possible interaction of SR-A and IRF5 and found that there is a possibility of two protein interactions (as follows). However, it needs further studied.

The software predicted that SR-A and IRF5 could be combined

1. Mori T, *et al.* Endocytic pathway of scavenger receptors via trans-Golgi system in bovine alveolar macrophages. *Laboratory investigation; a journal of technical methods and pathology* **71**, 409-416 (1994).
2. Tsay HJ, *et al.* Identifying N-linked glycan moiety and motifs in the cysteine-rich domain critical for N-glycosylation and intracellular trafficking of SR-AI and MARCO. *Journal of biomedical science* **23**, 27 (2016).
3. Yu X, *et al.* Pattern recognition scavenger receptor CD204 attenuates Toll-like receptor 4-induced NF-kappaB activation by directly inhibiting ubiquitination of tumor necrosis factor (TNF) receptor-associated factor 6. *The Journal of biological chemistry* **286**, 18795-18806 (2011).
4. Kiyonagi T, *et al.* Involvement of cholesterol-enriched microdomains in class A scavenger receptor-mediated responses in human macrophages. *Atherosclerosis* **215**, 60-69 (2011).

Reviewer 2 (Remarks to the Author):

The manuscript by Xu et al. is exciting because of novel data identifying a molecular mechanism for macrophage polarization and also because of new information about the role of alternatively activated macrophages in type 2 pathology. The data suggests that M2 polarization is critical for granuloma development and fibrosis in response to *S. japonicum* at the time point shown.

The authors support their conclusions effectively with well-controlled experiments. They show that macrophages from SR-A KO mice do not exhibit normal polarization to an M2 phenotype in the presence of type 2 stimuli. They explain that the immune dysregulation is due to SR-A from macrophages using adoptive transfer and siRNA transfection/overexpression approaches. They also use sh-IRF5 to demonstrate that IRF5 is necessary for the SR-A KO phenotype.

While I find the data convincing, I have a few comments and questions.

1. I think the observations about SR-A during *S. japonicum* infection themselves are interesting to immunology and parasitology communities, but the manuscript would be stronger if the authors added more context to their findings. The field still lacks good tools to study precise roles of M2 polarized macrophages. 1) It would be helpful if the authors explained how prevalent SR-A-expressing macrophages are. 2) Is the KO affecting a subset of M2 macrophages or all of them during *S. japonicum* infection? 3) Do the authors think there are SR-A/IRF5-independent mechanisms of polarization towards an M2 phenotype? 4) Showing this phenomenon is not *S. japonicum*-specific could also help to address this point.

Response: We appreciate the helpful comments of the reviewer.

1) Macrophages play major roles as sentinels for first line alerts and as mediators that shape the adaptive immune responses during infection. SR-A is constitutively expressed in most macrophages (not only limited to M1 and M2, but also BMDMs), in addition, the expression level of SRA could be regulated^{1, 2, 3, 4}. We have modified our Introduction section on page 4 lines 3-5.

2) As suggested by the reviewer, we have carried out additional experiments regarding M2 subsets. Results showed that *S. japonicum* infection significantly induced a M2b-dominant macrophages (TNF- α^{high} , IL-10^{high}, MR), but not M2a or M2c (TNF- α^{low} , IL-10^{low}, IL-6^{low}) in WT mice. Interestingly, SR-A-deficiency resulted in phenotypic M1-dominant (TNF- α^{high} ,

IL-10^{low}, MR^{low}, IL-12^{high}) macrophages present in the liver and peritoneal cavity during *S. japonicum* infection. We have added these results in our manuscript page 7 lines 16-20.

3) Yes, it is very likely that there exists the SR-A/IRF5-independent mechanism of polarization towards an M2 phenotype, as we had observed that the blocking of SR-A/IRF5 interaction (eg. through knockout of SR-A in Fig. 3) only resulted in a partial, not a complete inhibition of M2 differentiation. However, the underlying mechanism needs further study.

4) We had also found that SR-A-deficiency induced M1-dominant macrophages in heat-inactivated *M. tuberculosis* immunized mice (Supplementary Fig.3b). In addition, study shows that SR-A-deficiency also results in increased M1-dominant macrophages in myocardial infarction-induced cardiomyocyte necrosis model⁵. All these results suggest this phenomenon may not be *S. japonicum* specific.

1. de Villiers WJ, Fraser IP, Gordon S. Cytokine and growth factor regulation of macrophage scavenger receptor expression and function. *Immunology letters* **43**, 73-79 (1994).
2. Martinez FO, Gordon S, Locati M, Mantovani A. Transcriptional profiling of the human monocyte-to-macrophage differentiation and polarization: new molecules and patterns of gene expression. *J Immunol* **177**, 7303-7311 (2006).
3. Mukhopadhyay S, Gordon S. The role of scavenger receptors in pathogen recognition and innate immunity. *Immunobiology* **209**, 39-49 (2004).
4. Hoebe K, Janssen E, Beutler B. The interface between innate and adaptive immunity. *Nature immunology* **5**, 971-974 (2004).
5. Hu Y, *et al.* Class A scavenger receptor attenuates myocardial infarction-induced cardiomyocyte necrosis through suppressing M1 macrophage subset polarization. *Basic Res Cardiol* **106**, 1311-1328 (2011).

2. To add to the previous point, it would also be useful if the authors included a reference to Vannella et al. PLoS Pathog 10: e1004372 (2014) in the discussion. That publication described that removing IL-4R α signaling on resident macrophages resulted in a failure to downmodulate granuloma size in *S. mansoni* infection. It would be helpful for the authors to speculate about reasons for these different results.

Response: Our study suggested that the binding of SR-A to cytoplasmic IRF5 suppresses IRF5 nuclear translocation, which promotes a M2 polarization and increases IL-4 production in macrophages, then subsequently favors Th2 induction (Supplementary Fig. 9). In addition to our SR-A/IRF5 interaction, studies have demonstrated that except for IL-4R signaling, Jmjd3-Irf4 axis¹, Kruppel-like factor 4 (KLF4)², and PPAR γ ³ may also induce the M2 polarization of macrophages independent/dependent of IL-4. Vannella et al.'s results not only suggest that IL-4/IL-4R α signaling is not the only factor that induces M2 polarization, but

also support the novelty of our study that SR-A/IRF5 signaling is another factor that promotes M2 differentiation. As suggested by reviewer, we have cited this study and modified our discussion in our manuscript (page 18 lines 16-20).

1. Satoh T, *et al.* The Jmjd3-Irf4 axis regulates M2 macrophage polarization and host responses against helminth infection. *Nature immunology* **11**, 936-944 (2010).
2. Liao X, *et al.* Kruppel-like factor 4 regulates macrophage polarization. *The Journal of clinical investigation* **121**, 2736-2749 (2011).
3. Odegaard JI, *et al.* Macrophage-specific PPARgamma controls alternative activation and improves insulin resistance. *Nature* **447**, 1116-1120 (2007).

3. Krausgruber et al. *Nature Immunology* 12, 231-238 (2011) has previously shown the importance of IRF5 in macrophage polarization so it warrants more of a discussion as well to put the new findings into context.

Response: We have followed the reviewer's suggestion to add this in the discussion (page 18 lines 12-15) as following: Krausgruber *et al.* found the importance of IRF5 in the regulation of macrophage polarization, however, the precise mechanism is unknown. Our study has further shed light on significance of SR-A/IRF5 interaction to regulate macrophage polarization and Th1/Th2 differentiation.

4. The authors should explain why they used CD16/32 as a marker of M1 cells. Did its expression track with MHC class II or CD80/86? It would also be helpful for the authors to include a diagram of flow plots as an example of the macrophage gating strategy.

Response:

M1 macrophages highly express CD16/32, MHC class II, and CD80/86, and CD16/32 is one of well accepted M1 makers according to previous literatures^{1,2,3}. In our study, the expression of CD16/32 was consistent with MHC class II or CD80/86 after in vitro stimulation with SEA (Supplementary Fig.3b).

Following the reviewer's suggestion, we have added the macrophage gating strategy in Supplementary Fig.3a

1. Kroner A, Greenhalgh AD, Zarruk JG, Passos Dos Santos R, Gaestel M, David S. TNF and increased intracellular iron alter macrophage polarization to a detrimental M1 phenotype in the injured spinal cord. *Neuron* **83**, 1098-1116 (2014).
2. Zhu W, *et al.* Disequilibrium of M1 and M2 macrophages correlates with the development of experimental inflammatory bowel diseases. *Immunological investigations* **43**, 638-652 (2014).
3. Willemen HL, Huo XJ, Mao-Ying QL, Zijlstra J, Heijnen CJ, Kavelaars A. MicroRNA-124 as a novel treatment for persistent hyperalgesia. *Journal of neuroinflammation* **9**, 143 (2012).

5. The results/methods/figure legends should be clarified in some spots. For example, it is not clear what time-point is used for the pathology shown in Figure 6. In the Figure 1 legend, the micrographs of intestine tissue are not labeled. These changes will help make the experiments more reproducible and interpretable.

Response: Thanks so much for the reviewer's suggestion, as suggested by the reviewer, we have extensively revised our manuscript to make it more clarified. We have added the time-point in Figure 6 in our manuscript page 45 lines 4-6 as follows: "Mice were intravenously injected with shRNA lentiviral particles targeting IRF5 since 3.5 weeks post *S. japonicum* infection by weekly for 4 weeks and sacrificed at 7.5 weeks post *S. japonicum* infection for further study". We have also labeled the micrographs of intestine tissue in Figure 1 legend.

Reviewers' comments:

Reviewer #1 (Remarks to the Author):

In this revised manuscript, the authors have done a reasonable effort in revising their manuscript. Most of the comments were addressed as far as is possible. However, some additional comments are:

1. pg. 7 The use of M2a, M2b etc. must be removed - there is no such strict definition and this nomenclature is not accepted in the field. Please refer to Murray et al. *Immunity*, 2014 for the reasoning, or Murray *Ann Rev. Physiol.* 2017 for an updated appraisal of this issue.

In my view the figures are too small and crowded. The figures must be revised to be legible and clear. This is especially the case of Figure 1 which has the most important data – the microscopy should be large and crystal clear.

As regards the data concerning IRF5, my (strong) suggestion to the authors and Editors, is that:

- (i) The legend and description of figure 5 should be changed to reflect that increased IRF5 amounts were associated with the phenotype in the SR-A mice. 'Associated' is the key word here, as in my view much more work needs to be done on this association.
- (ii) In Figure 6, f-h should be made larger and a-e moved to the supplemental data. However, an essential experimental addition to figure 6 would be gel images showing a clear depletion of IFR5 in both WT and KO model cells. shRNA data gathered from in vivo systems is often questionable – correct and robust controls must be shown.
- (iii) Figure 7 should be revised and minimized to include as part of figure 6. The authors should only show the microscopy and one (the best) of the ci-IPs and then state that the results are consistent with a potential interplay. They could then speculate further in the discussion. As presented, the conclusion is too assertive and would be met with skepticism by the field.

Reviewer #2 (Remarks to the Author):

I think the authors have satisfactorily addressed the points I raised. While I find the results to be surprising and novel, the data and methods look sound to me.

The additional adoptive transfer experiment (Supp Fig 4g-n) where WT or KO macrophages were transferred into infected mice is a key addition. It supplies evidence that SR-A on macrophages and not SR-A on other cell types is critical.

There remain some outstanding issues that the authors should emphasize in the text. These will help readers interpret the results in the context of existing literature:

- 1) SR-A has been shown to be expressed by cell types other than macrophages. It is important to cite literature like Kelley et al. (2014) *Crit Rev Immunol* 34 that lists the other cell types, and to explain this is why the new adoptive transfer experiments are important for the conclusions of the paper. Interpretations of the disease phenotype in SR-A KO mice should also be written with this in mind.
- 2) I'd recommend moving the results from the new adoptive transfer experiment into the main figures. Perhaps current main Figures 1 and 2 can be combined to make space.
- 3) The data is not consistent with some past studies that found SR-A and M2 macs can functionally limit inflammation (Arredouani et al. (2007) *J Immunol* 178 in a OVA allergy model and Vannella et al (2014) *PLoS Pathog* 10 in *S. mansoni* infection, respectively, come to mind). The authors find the opposite result. The Arredoani study could be different because DCs are less influential in the

liver like the authors already note in the Discussion. I asked the authors to include the Vannella paper for context in the first round of review, but the authors misinterpreted the findings. That paper does not necessarily suggest there are other means of M2 polarization beyond the IL-4Ra. Instead the authors should try to explain why M2 macrophages seem to drive granulomatous inflammation in their paper, but M2 macrophages seem to limit granulomatous inflammation in the Vannella paper.

We appreciate the helpful and constructive comments of the reviewers again. We have carried out the additional experiments suggested by the reviewers and added the new data into figure (Supplementary Fig.7b). The newly acquired results further supported the conclusions described in the manuscript. We have also extensively revised our manuscript.

Our point-by-point responses to the reviewers' comments are described below:

Reviewer #1 (Remarks to the Author):

In this revised manuscript, the authors have done a reasonable effort in revising their manuscript. Most of the comments were addressed as far as is possible. However, some additional comments are:

1. pg. 7 The use of M2a, M2b etc. must be removed - there is no such strict definition and this nomenclature is not accepted in the field. Please refer to Murray et al. Immunity, 2014 for the reasoning, or Murray Ann Rev. Physiol. 2017 for an updated appraisal of this issue.

Response: We appreciate the helpful suggestion of the reviewer and we have deleted the results and description of M2a and M2b.

In my view the figures are too small and crowded. The figures must be revised to be legible and clear. This is especially the case of Figure 1 which has the most important data – the microscopy should be large and crystal clear.

Response: According to the submission requirements for peer review of Nature Communications, we have to put the Figures in the word file, which caused figures too small and crowded. In fact, our original figures, especially .eps format diagram by Adobe Illustrator CS4 software, are much better for possible publication.

As regards the data concerning IRF5, my (strong) suggestion to the authors and Editors, is that:

(i) The legend and description of figure 5 should be changed to reflect that increased IRF5 amounts were associated with the phenotype in the SR-A mice. 'Associated' is the key word here, as in my view much more work needs to be done on this association.

Response: According to the reviewer's suggestion, we have re-written the legend and description of figure 5 as follows: "SR-A-promoted M2 polarization is associated with decreased IRF5 amounts" on page 42 lines 2-3 and "SR-A-promoted M2 polarization together with the resultant CD4⁺T cell differentiation are associated with decreased IRF5 amounts" on page 9 lines 20-21.

(ii) In Figure 6, f-h should be made larger and a-e moved to the supplemental data. However, an essential experimental addition to figure 6 would be gel images showing a clear depletion of IRF5 in both WT and KO model cells. shRNA data gathered from in vivo systems is often questionable – correct and robust controls must be shown.

Response: As suggested by the reviewer, we have moved Fig 6a-e to the supplemental data and revised f-h to be large. We have also added the levels of IRF5 in both infected WT and infected KO mice in Fig S7b as an experimental addition to figure 6, and found that the levels of IRF5 were decreased in the livers of infected WT and infected KO mice.

(iii) Figure 7 should be revised and minimized to include as part of figure 6. The authors should only show the microscopy and one (the best) of the co-IPs and then state that the results are consistent with a potential interplay. They could then speculate further in the discussion. As presented, the conclusion is too assertive and would be met with skepticism by the field.

Response: We have followed the reviewer's suggestion to revise Figure 7 and minimized it to include as part of figure 6d-g. We also have modified our description and conclusion as requested by the reviewer in our revised manuscript in page 17 lines 11-14.

Reviewer #2 (Remarks to the Author):

I think the authors have satisfactorily addressed the points I raised. While I find the results to be surprising and novel, the data and methods look sound to me.

The additional adoptive transfer experiment (Supp Fig 4g-n) where WT or KO macrophages were transferred into infected mice is a key addition. It supplies evidence that SR-A on macrophages and not SR-A on other cell types is critical.

There remain some outstanding issues that the authors should emphasize in the text. These will help readers interpret the results in the context of existing literature:

1) SR-A has been shown to be expressed by cell types other than macrophages. It is important to cite literature like Kelley et al. (2014) *Crit Rev Immunol* 34 that lists the other cell types, and to explain this is why the new adoptive transfer experiments are important for the conclusions of the paper. Interpretations of the disease phenotype in SR-A KO mice should also be written with this in mind.

Response: We have cited this study and modified our discussion in our manuscript (page 15 lines 13-17) as follows: SR-A has been shown to be also expressed by vascular smooth muscle cells, endothelial cells, human lung epithelial cells, and microglia, etc.¹, however, SR-A is predominantly expressed in macrophages, which plays a much more important role than other cells mentioned above in schistosomiasis²⁻⁴. Therefore, we used adoptive transfer experiment to better investigate the role of macrophage in the regulation of CD4⁺T cell differentiation and granuloma formation.

- 1 Kelley, J. L., Ozment, T. R., Li, C., Schweitzer, J. B. & Williams, D. L. Scavenger receptor-A (CD204): a two-edged sword in health and disease. *Critical reviews in immunology* **34**, 241-261 (2014).
- 2 Pearce, E. J. & MacDonald, A. S. The immunobiology of schistosomiasis. *Nature reviews. Immunology* **2**, 499-511, doi:10.1038/nri843 (2002).
- 3 Barron, L. & Wynn, T. A. Macrophage activation governs schistosomiasis-induced inflammation and fibrosis. *European Journal of Immunology* **41**, 2509-2514, doi:10.1002/eji.201141869 (2011).
- 4 Herbert, D. R. *et al.* Arginase I suppresses IL-12/IL-23p40-driven intestinal inflammation during acute schistosomiasis. *J Immunol* **184**, 6438-6446, doi:10.4049/jimmunol.0902009 (2010).

2) I'd recommend moving the results from the new adoptive transfer experiment into the main figures. Perhaps current main Figures 1 and 2 can be combined to make space.

Response: We have followed the reviewer's suggestion and moved Supplementary Fig 4g-n to Fig 4d-g.

3) The data is not consistent with some past studies that found SR-A and M2 macs can functionally limit inflammation (Arredouani et al. (2007) *J Immunol* 178 in a OVA allergy model and Vannella et al (2014) *PLoS Pathog* 10 in *S. mansoni* infection, respectively, come to mind). The authors find the opposite result. The Arredoani study could be different because DCs are less influential in the liver like the authors already note in the Discussion. I asked the authors to include the Vannella paper for context in the first round of review, but the authors misinterpreted the findings. That paper does not necessarily suggest there are other means of M2 polarization beyond the IL-4Ra. Instead the authors should try to explain why M2 macrophages seem to drive granulomatous inflammation in their paper, but M2 macrophages seem to limit granulomatous inflammation in the Vannella paper.

Response: We really appreciate for the reviewer's patience and advice. Vannella et al. found that removing IL-4Ralpha signaling on resident macrophages resulted in a failure to downmodulate granuloma size in *S. mansoni* infection. However, there are also many previous works, including IL-4R^{-/-} mice used in Reena Rani et al's study, supporting M2 macrophages contribute to the granuloma pathology during schistosoma infection¹⁻⁵, although currently we don't know the exact underlining mechanisms. IL-4R^{flox/Δ}LysM^{Cre} mice in Vannella et al's study displayed no reduction in the expression of multiple genes that characterize the alternatively-activated macrophage (AAM, also called M2) phenotype including Ym1, Retnla (Relm-a), and Arg1. However, SR-A^{-/-} mice used in our study showed significantly less changes in M2-related genes (Ym1, Arg1 etc.) and decreased granuloma size. Considering the big differences in consequence and phenotype after IL-4R or SR-A deficiency, or maybe any other unknown factors, we have to remove the inappropriate citation and discussion on page 17 line 18-22 and page 18 line 1-2.

- 1 Rani, R., Jordan, M. B., Divanovic, S. & Herbert, D. R. IFN-gamma-driven IDO production from macrophages protects IL-4Ralpha-deficient mice against lethality during *Schistosoma mansoni* infection. *The American journal of pathology* **180**, 2001-2008, doi:10.1016/j.ajpath.2012.01.013 (2012).
- 2 Kreider, T., Anthony, R. M., Urban, J. F., Jr. & Gause, W. C. Alternatively activated macrophages in helminth infections. *Current opinion in immunology* **19**, 448-453, doi:10.1016/j.coi.2007.07.002 (2007).
- 3 Hesse, M., Cheever, A. W., Jankovic, D. & Wynn, T. A. NOS-2 mediates the protective anti-inflammatory and antifibrotic effects of the Th1-inducing adjuvant, IL-12, in a Th2 model of granulomatous disease. *The American journal of pathology* **157**, 945-955, doi:10.1016/S0002-9440(10)64607-X (2000).
- 4 Noel, W., Raes, G., Hassanzadeh Ghassabeh, G., De Baetselier, P. & Beschin, A. Alternatively activated macrophages during parasite infections. *Trends in parasitology* **20**, 126-133, doi:10.1016/j.pt.2004.01.004 (2004).
- 5 Hesse, M. *et al.* Differential regulation of nitric oxide synthase-2 and arginase-1 by type 1/type 2 cytokines in vivo: granulomatous pathology is shaped by the pattern of L-arginine metabolism. *J Immunol* **167**, 6533-6544 (2001).